# PLAN THEN ACT: BI-LEVEL CAD COMMAND SEQUENCE GENERATION

**Qiangya Guo**[1,*] **Gang Dai**[1*], **Zhuoman Liu**[3], **Shuangping Huang**[1,†],
**Yunqing Hu**[4], **Huiyuan Zhang**[4], **Tianshui Chen** [2]

[1]South China University of Technology
[2]Guangdong University of Technology
[3]Hong Kong Polytechnic University
[4]Zhuzhou CRRC Times Electric Co., Ltd.
`eegqy@mail.scut.edu.cn, daigang@gdut.edu.cn, eehsp@scut.edu.cn`

## ABSTRACT

Computer-Aided Design (CAD), renowned for its flexibility and precision, serves as the foundation of digital design. Recently, some efforts adopt Large Language Models (LLMs) for generating parametric CAD command sequences from text instructions. However, our study reveals that LLMs pre-trained on large-scale general data are not proficient at directly outputting task-specific CAD sequences. Instead of relying on direct generation, we introduce a *Plan then Act* process where user instructions are first parsed into a chain-like operational plan via an LLM, which is then used to generate accurate command sequences. Specifically, we propose PTA, a new bi-level CAD command sequence generation method. The PTA consists of two critical stages: high-level plan generation and low-level command generation. During the high-level stage, an LLM-based Planner completes the planning process, parsing user instructions into a high-level operation plan. Following this, at the low-level generation stage, we introduce an Actioner equipped with a requirement-aware mechanism to extract design requirements (*e.g.*, dimensions, geometric relationships) from user instructions. This extracted information is used to guide the low-level command sequence generation, improving the alignment of the generated sequences with user requirements. Experimental results demonstrate that our PTA outperforms existing methods in both quantitative and qualitative evaluations. Code is available at `https://github.com/QiferG/Plan-then-Act`.

## 1 INTRODUCTION

As a core technology in digital design, Computer-Aided Design (CAD) plays an indispensable role in industrial design (Cherng et al., 1998; Brozovsky et al., 2024). The goal of CAD is to produce CAD models based on user requirements (Miller, 2004; Ralph et al., 2009), requiring designers to leverage professional expertise in determining structural features and shape parameters. While current CAD software offers extensive design functions, two practical challenges persist: 1) designers must master specialized skills, such as parameter configuration and geometric constraint definition; 2) the design process involves numerous operations, which remain time-consuming and inefficient.

Recently, Text-based CAD generation attracts significant attention (Xu et al., 2024; Alrashedy et al., 2024; Wang et al., 2025b). For instance, Khan et al. (2024a) and Li et al. (2024b) use text encoders to interpret user instructions and autoregressively generate CAD sequences. These methods (Khan et al., 2024a; Li et al., 2024b)trained from scratch have limited semantic parsing and modeling capabilities and may struggle with complex CAD modeling tasks. Therefore, some methods (Xu et al., 2024; Li et al., 2025a; Wang et al., 2025b) attempt to use large language models (LLMs) pre-trained on large-scale general data to understand user input and generate CAD command sequences.

---

*Authors contributed equally.
†Corresponding author

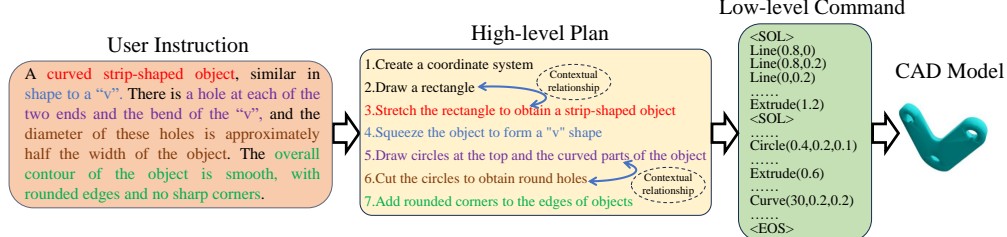

Figure 1: Illustration of a user instruction, high-level plan, low-level command sequence, and CAD model result. The same color represents operation steps and their most relevant requirement information in user instruction. The "Contextual relationship" indicates there is a strong contextual connection between the operation steps.

Although these LLM-based methods make some progress, our investigation finds that LLMs are not proficient at directly outputting task-specific CAD control sequences (cf. Section 3).

To address the above issues, inspired by human planning strategies (Masicampo & Baumeister, 2011; Rogers et al., 2013; 2015), we propose a reasonable assumption that decomposing abstract user instructions into a chain-like operation plan can improve the accuracy of command sequence generation. To validate this assumption, we compare the performance of CAD generation using user instructions alone versus combining them with operation plans. The results (cf. Section 3) demonstrate the effectiveness of the chain-like operation plan. Based on the above analysis, we do not request LLMs to generate command sequences directly. Instead, we consider a process where user instructions are first parsed into a chain-like operational plan via an LLM, followed by the generation of the accurate command sequence.

In this work, we propose PTA (Plan then Act), a new bi-level CAD command sequence generation method. The PTA consists of two key stages: high-level plan generation and low-level command sequence generation. During the high-level stage, we develop a Qwen3-8B (Yang et al., 2025a) based Planner to translate abstract user instructions into a chain-like operation plan. While at the low-level stage, we design an Actioner equipped with a requirement-aware mechanism to generate accurate low-level executable command sequences.

For the design of Actioner, we consider that while the high-level plan is derived from the user instruction, it primarily provides global operational guidance for command sequence generation. To produce an accurate and precise command sequence, it is essential to integrate the high-level plan with detailed operational specifications. As shown in Figure 1, we focus on two aspects: 1) When executing each operation step in a high-level plan, it should not equally focus on all information in the user instructions, instead, it should pay more attention on the most relevant requirement information for current operation step; 2) The operation steps are not isolated but have contextual relationships with each other. Based on the above considerations, we design the requirement-aware mechanism. First, we use cross-attention, employing the operation steps in the high-level plan as queries to retrieve the most relevant requirement information. Then, we use self-attention to capture the contextual relationships between the operational steps. The final fused information guides the Transformer decoder to generate an accurate low-level CAD command sequence. This awareness and integration of requirement information enhances the alignment of the generated command sequence with user requirements.

We summarize our contributions as follows:

- We propose PTA, a new bi-level CAD modeling method that first decomposes user instructions into high-level operation plans via LLM, then completes the low-level command sequence generation.

- We design a requirement-aware mechanism that automatically extracts critical requirement information from the user instruction, improving the alignment of the generated CAD sequences with user requirements.

- Experimental evaluations demonstrate that our PTA achieves superior performance compared to available state-of-the-art methods.

## 2 REALTED WORK

### 2.1 CAD GENERATION

CAD software typically represents CAD models as Boundary Representation (B-Rep) (Lambourne et al., 2021), which defines the model's boundary using geometric elements (e.g., point, curve, surface) and their topological relationships. Pie-net (Wang et al., 2020) and ComplexGen (Guo et al., 2022) generate B-rep data by predicting point cloud edges, while Solidgen (Jayaraman et al., 2022) employs a pointer network to produce points and faces, constructing a complete B-rep topology. However, B-Rep only stores geometric appearance, ignoring the operation process. DeepCAD (Wu et al., 2021)introduces a command sequence representation, expressing the CAD model as a chain of 2D sketches (*e.g.*, circles, lines, arcs) and 3D operations (*e.g.*, extrusion). This sequence-based representation enhances the interpretability and editability of CAD generation and supports designers in flexibly modifying operation steps (Khan et al., 2024b). Moreover, some studies (Badagabettu et al., 2024; Alrashedy et al., 2024; Wang et al., 2025c; Du et al., 2024) explore CAD code generation, obtaining CAD models by executing generated code.

Wu et al. (2021); Xu et al. (2022; 2023); Li et al. (2025b;c) encode CAD models into high-dimensional vectors or codebooks, and generate command sequences based on high-dimensional representations. These data, such as vectors or codebooks, are abstract and not convenient for user interaction. Li et al. (2022); Wang et al. (2025c); Qin et al. (2025); Karadeniz et al. (2025); Mallis et al. (2024); Sanghi et al. (2023) utilizes sketches to drive the generation of CAD command sequences. However, sketch data requires professional designers to draw, making it unfriendly for CAD beginners. Some studies (Li et al., 2024a; Dupont et al., 2022; Lambourne et al., 2022; Li et al., 2023) focus on CAD reverse engineering. You et al. (2024); Chen et al. (2025b;a) generates CAD operation sequences from CAD images, while Yang et al. (2025b); Khan et al. (2024a); Ma et al. (2024) focuses on point cloud-driven CAD generation. Yin et al. (2025); Zhou et al. (2023); Xu et al. (2021); Dupont et al. (2024), on the other hand, considers using B-rep as conditional data to generate the corresponding CAD operation sequences.

Natural language, characterized by high flexibility and accessibility, is commonly used to express requirements. Some research (Khan et al., 2024b; Li et al., 2024b) aims to use natural language instructions to drive the generation of CAD command sequences. Text2CAD (Khan et al., 2024b) first proposes an end-to-end generation framework that translates text instructions into CAD command sequences. CAD Translator (Li et al., 2024b) proposes a cascading contrastive strategy to align text with CAD sequence. Many works (Li et al., 2025a; Liao et al., 2025; Guan et al., 2025; Wang et al., 2025a; Gong et al.)explore how LLMs empower natural language text instruction to generate CAD models. (Zhang et al., 2024; Wu et al., 2024; Yuan et al., 2025; Zhang et al., 2025) fine-tune Multimodal Large Language Models (MLLMs) by taking basic CAD models and editing control conditions, enabling modifications or completions of CAD models. Xu et al. (2024); Kolodiazhnyi et al. (2025) utilizes MLLMs to generate command sequences from multimodal data (*i.e.*, text, images, and point clouds). Different from these methods that request LLMs to generate command sequences directly, we consider a process where user instructions are first parsed into an operation plan, followed by the generation of the command sequence.

## 3 MOTIVATION

In this section, we conduct detailed analyzes of the above-discussed points: 1) pre-trained LLMs on large-scale general data struggle to directly generate task-specific low-level CAD control commands; 2) operation plans can improve the accuracy of generating CAD command sequences.

We first explore scenarios where LLMs are used to directly generate low-level CAD command sequences. We select the state-of-the-art closed-source LLM GPT-4o (Hurst et al., 2024) and the open-source LLMs LLaMA3.1-8B (Dubey et al., 2024) and Qwen3-8B (Yang et al., 2025a). For GPT-4o, we provide an example of a user instruction-command sequence and prompt it to generate command sequences based on user instructions. Unfortunately, the Invalid rate (IR) of the CAD sequences generated by GPT-4o reached 70.35%. This demonstrates that the sequences generated by GPT-4o mostly cannot be correctly executed to produce 3D CAD models. We also fine-tune LLaMA3.1-8B and Qwen3-8B to enhance their ability to generate CAD command sequences. However, despite some performance improvements, they still achieve Invalid rates of 33.46% and

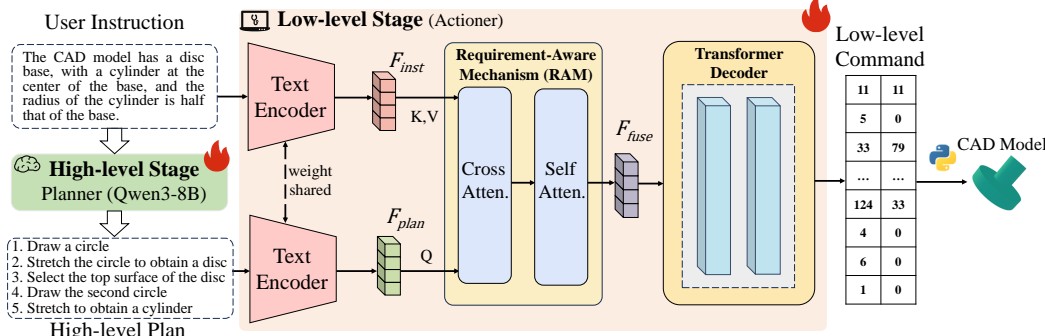

Figure 2: Overview of the proposed PTA. The PTA comprises two stages: the high-level plan generation and the low-level command sequence generation. During the high-level stage, a Planner parses user instructions into a high-level operation plan. While at the low-level stage, an Actioner maps the high-level plan into a low-level executable command sequence. Within the Actioner, the user instruction and high-level plan are separately fed into a text encoder to obtain instruction features $F_{inst}$ and plan features $F_{plan}$. Subsequently, $F_{inst}$ and $F_{plan}$ are used as inputs to a Requirement-Aware Mechanism(RAM). The RAM employs $F_{plan}$ as the query to automatically perceive and fuse the most critical design requirements from user instructions. The fused features $F_{fuse}$ serve as guidance for the Transformer decoder to direct the generation of a low-level command sequence.

20.06%, respectively. Furthermore, we apply direct preference optimization (DPO) (Rafailov et al., 2023) to the finetuned Qwen3-8B. However, the invalid rate still reached 18.29%. The high Invalid rates indicate that LLMs struggle to learn the low-level CAD control command chains. For details of the preliminary experiments, please refer to the Appendix A.4.

Next, we analyze the assumption that "operation plans can improve the accuracy of CAD sequences generation" by comparing two scenarios: 1) generating command sequences based on only user instructions, and 2) incorporating operation plans into user instructions. We conduct experiments on a task-specific model trained from scratch and find that both scenarios achieve low invalid rates(2.66% and 0.63%), but with the operation plan included, the generated command sequences show significant improvement in metrics: Median CD (Chamfer Distance) improved from 200.32 to 104.56, and F1 score increased from 45.16 to 67.42. These results validate our assumption. For details regarding the Invalid Rate (IR), Chamfer Distance (CD), F1 score, please refer to Section 5.1.3.

In practice, to avoid increasing the user's input burden, the operation plan should be automatically derived by the model through parsing user instructions. Although LLMs struggle to directly generate low-level command sequences, we aim to leverage their semantic parsing and reasoning abilities to generate operation plans, improving the accuracy of command sequence generation.

# 4 METHOD

## 4.1 OVERALL SCHEME

Our core idea is to use LLM to parse user instructions into operation plans, improving the accuracy of CAD sequence generation. We propose PTA, a new bi-level generation method. As shown in Figure 2, our PTA comprises two stages: high-level plan generation and low-level command sequence generation. First, during the high-level generation stage, we fine-tune Qwen3-8B (Yang et al., 2025a) to serve as a Planner. The Planner's role is to analyze user intent and generate a high-level operation plan. This stage is defined as:

$$P = Planner(I), \tag{1}$$

where $P$ denotes high-level plan and $I$ represents user instruction.

Subsequently, at the low-level command sequence generation stage, we input the original user instruction and generated high-level plan into an Actioner equipped with a requirement-aware mechanism. This mechanism uses the operation steps in the high-level plan as queries to automatically perceive and fuse requirement information from the user instruction. The fused information serves

**Prompt:** you are a CAD task planner. Your role is to generate CAD operation steps as a high-level operation plan based on user design instruction. There is an example for reference.

  example: { user instruction :" A rectangular prism with a flat top and bottom. The length of the prism is approximately three times its width. The height of the prism is approximately two times its width. The prism is solid and has no holes or openings. ",

  high-level plan:"

  1. Create the coordinate system

  2. Draw a rectangle

  4. scale the rectangle

  5. Transform the 2D sketch into 3D

  6. Extrude the 2D sketch to get a rectangular prism "}

  Now, based on the given reference, parse user instructions into a high-level operation plan. Output only the high-level plan.

Figure 3: An example prompt is provided for the Planner.

as contextual guidance, directing the Transformer decoder to generate a low-level executable command sequence. We define the low-level generation stage as:

$$\hat{C} = Actioner(I, P), \tag{2}$$

where $\hat{C}$ denotes the low-level command sequence.

## 4.2 HIGH-LEVEL PLAN GENERATION

In the high-level plan generation stage, our goal is to parse CAD design tasks into operation plan via an LLM. Although LLMs demonstrate strong semantic understanding and reasoning capabilities, some studies (Yao et al., 2025; Liu et al., 2025) point out that hallucination in LLMs can lead to erroneous information during reasoning. Based on this consideration, we do not request the LLM to fully parse user requirements and directly generate an operational process with detailed operation types and specific parameters. Instead, we aim to analyze user intent and provide a relatively simple yet clear high-level process plan.

In practice, we select the open-source LLM Qwen3-8B (Yang et al., 2025a) to be a base Planner. As shown in Figure 3, we use a prompt that includes an example to guide Qwen3-8B to assume the role of a CAD Planner, encouraging it to generate high-level operation plans. To enhance its professional capabilities, we perform full-parameter supervised fine-tuning (SFT) using paired user instruction $I$ and high-level plan $P$, along with the prompt (the details of the training process and dataset are provided in Section 5.1.2 and Appendix A.2). The process is supervised by an objective $\mathcal{L}_{plan}$:

$$\mathcal{L}_{plan} = -\sum_{t=1}^{T} \log P(p_t \mid p_{<t}, I, Prompt), \tag{3}$$

where $p_t$ denotes the token at time step $t$ of the high-level plan.

## 4.3 LOW-LEVEL COMMAND SEQUENCE GENERATION

At this stage, our goal is to efficiently utilize operation information in the high-level plan and the requirement information from user instructions to guide the generation of low-level executable command sequences. We design an Actioner equipped with a requirement-aware mechanism. As shown in Figure 2, within the Actioner, we employ a pre-trained BERT (Bidirectional Encoder Representations from Transformers) (Devlin et al., 2019) model as our text encoder. The user instruction $I$ and the high-level plan $P$ are separately fed into the text encoder to obtain instruction features $F_{inst} \in \mathbb{R}^{N_p \times d_p}$ and plan features $F_{plan} \in \mathbb{R}^{N_p \times d_p}$, $N_p$ is the length of the text encoding, and $d_p$ is the dimension of the encoding:

$$F_{inst} = BERT(I), \tag{4}$$
$$F_{plan} = BERT(P). \tag{5}$$

Then, a requirement-aware mechanism (RAM) is designed to extract requirement information from $F_{inst}$ and integrate it with $F_{plan}$. The fused features $F_{fuse}$ guide a Transformer decoder to generate CAD command sequences.

### 4.3.1 REQUIREMENT-AWARE MECHANISM (RAM)

To effectively utilize requirement information provided in user instructions, we introduce a Requirement-Aware Mechanism (RAM), as illustrated in Figure 2. This mechanism integrates in-

struction feature $F_{inst}$ and plan feature $F_{plan}$ into two multi-head attention modules. First, a cross-attention module uses the plan features as queries to identify the most critical requirement from the user instructions for operation steps. This process is represented as:

$$O = Atten_1(Q_1 = F_{plan}, K_1 = V_1 = F_{inst}). \tag{6}$$

Next, we simply add the output $O$ and plan features $F_{plan}$ to obtain a preliminary fused vector. Further, considering the contextual relationships among different operations in the plan, we employ this preliminary fused vector as the query, key, and value into a self-attention module to obtain the final, fully integrated vector. This fused vector serves as guidance information to instruct the Transformer decoder in generating the final low-level executable command sequence. The self-attention is defined as:

$$F_{fuse} = Atten_2(Q_2 = K_2 = V_2 = F_{plan} + O). \tag{7}$$

### 4.3.2 TRANSFORMER DECODER FOR CAD SEQUENCE

Following (Wu et al., 2021; Khan et al., 2024a;b), we represent the CAD command sequence as a chain of sketch primitives and extrusion operations. Each token in the command sequence is a 2D token, representing either the 2D-coordinates of sketch primitives, extrusion operation parameters (*e.g.*, euler angles, extrusion distance, Boolean operations, sketch scale), or end markers(*e.g.*, curve/loop/sketch/end sequence). We initialize tokens in the CAD command sequence as one-hot vectors with a dimension of 256. The input command sequence at timestep $t-1$ can be represented as $C_{1:t-1} \in \mathbb{R}^{N_{t-1} \times 2 \times 256}$. We simply split $C_{1:t-1}$ to $C^x_{1:t-1}$ and $C^y_{1:t-1} \in \mathbb{R}^{N_{t-1} \times 256}$. The input command sequence $F^0_{t-1} \in \mathbb{R}^{N_{t-1} \times d}$ embedding can be obtained via Equation 8:

$$F^0_{t-1} = C^x_{1:t-1} W^x_{t-1} + C^y_{1:t-1} W^y_{t-1} + pos, \tag{8}$$

where $W^x_{t-1}, W^y_{t-1} \in \mathbb{R}^{N_{t-1} \times d}$ are learnable weight, $pos \in \mathbb{R}^{N_{t-1} \times d}$ is the positional embedding.

We utilize a Transformer decoder (Vaswani et al., 2017; Dai et al., 2023) to fuse the guidance vector $F_{fuse}$ and generate the executable command sequence. The decoder consists of $L$ blocks; each block $l$ takes as input the command sequence embedding $F^{l-1}_{t-1}$ output from the previous block, along with the guidance vector $F_{fuse}$. First, $F^{l-1}_{t-1}$ undergoes a self-attention operation. And then it serves as a query to retrieve relevant information from the fused guidance vector $F_{fuse}$, generating a new command sequence embedding $F^l_{t-1}$. Finally, the embedding output from the last layer passes through an MLP to produce the final executable command sequence. The training process is supervised by the objective $\mathcal{L}_{act}$:

$$\mathcal{L}_{act} = -\sum_{t=1}^{N_c} \log P(c_t \mid c_{<t}, I, P), \tag{9}$$

where $c_t$ denotes token at time step $t$ of command sequence.

During inference, the input CAD command sequence is unavailable. Instead, we use the generated tokens $\{\hat{c}_j\}_{j=1}^{t-1}$ as the input of the step $t$, combine them with $F_{fuse}$ to predict the next token. It is necessary to emphasize that during the training process of the Actioner, real high-level plan data is required. However, during the inference process, the high-level operation plan is generated by the finetuned Planner based on user instructions rather than provided by the user, thus not imposing additional input pressure on users.

## 5 EXPERIMENTS

### 5.1 EXPERIMENTAL SETTINGS

#### 5.1.1 DATASET

We conduct experiments on the publicly available dataset Text2CAD (Khan et al., 2024b). Text2CAD dataset (Khan et al., 2024b) contains 170k CAD command sequences. Each CAD sequence corresponds to user instructions at four levels (*i.e.*, 0, 1, 2, 3), totaling 660k text instructions.

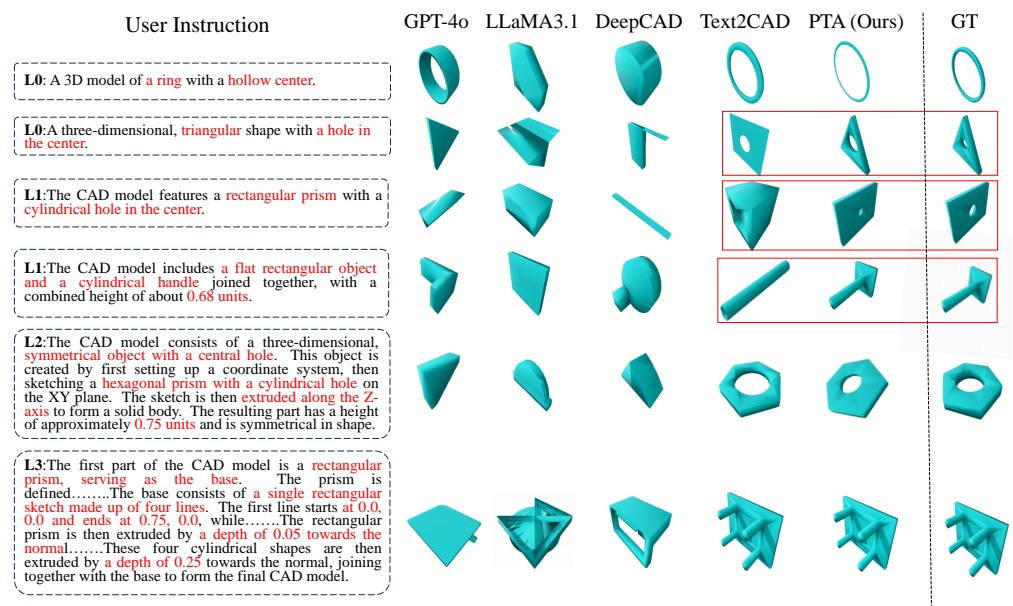

Figure 4: Qualitative comparisons between our method and SOTA CAD command sequence methods. Bold in "User Instruction" indicates the user instruction level, while red highlights shapes or details that require more attention. Due to the L3 Instruction being too long, we make some omissions. The red boxes specifically compare Text2CAD and our PTA. In the second row, our PTA follows the user instructions to create a triangle, while Text2CAD incorrectly generates a rectangle. Additional visual results are provided in the Appendix A.7.

Following the original Text2CAD dataset split, we use 150k CAD sequences for training and 8k test and validation sequences.

In addition to the pairs of (user instruction, command sequence), the Text2CAD dataset (Khan et al., 2024b) also provides nli_data. The nli_data can be regarded as a natural language translation of the CAD command sequence, containing the most detailed low-level operations and specific operational values. In this paper, we prompt Qwen2.5-32B-Instruct Qwen et al. (2025) to extract or aggregate a high-level operation plan from the nli_data (we provide the complete high-level construction process and some discussions in the Appendix A.2 and Figure 7).

### 5.1.2 IMPLEMENTATION DETAILS

PTA includes Planner and Actioner. For Planner, we choose Qwen3-8B (Yang et al., 2025a) as the base model, utilizing full parameter supervised fine-tuning. The maximum input sequence length is 2048. We use AdamW optimizer (Loshchilov & Hutter, 2017) with a learning rate of $10^{-4}$. The Planner is fine-tuned for 1 epoch on 8 NVIDIA RTX A800 GPUs (batch size of 6). For Actioner, the BERT (Devlin et al., 2019) encoder has an encoding dimension of 1024, with a maximum input sequence length of 512. The requirement-aware mechanism consists of three layers of cross-attention and three layers of self-attention, with 8 attention heads and a feature dimension of 1024. The Transformer decoder (Vaswani et al., 2017) includes 8 decoder blocks with 8 attention heads. The maximum sequence length for CAD is 272, with an encoding dimension of 256. During the optimization process, we freeze the parameters of the BERT encoder, using Adam optimizer (Kingma & Ba, 2014) with a learning rate of $10^{-4}$, and train for 40 epochs on 3 NVIDIA RTX 3090 GPUs with a batch size of 16.

### 5.1.3 EVALUATIONS

Following Text2CAD (Khan et al., 2024b), our metrics include: 1) GPT-4V evaluation, where GPT-4V (Achiam et al., 2023) is required to select the CAD model that best matches user instructions from the candidates generated by various methods; 2) using Chamfer Distance (CD) to assess the geometric alignment between the ground truth and predicted CAD models; 3) evaluating the F1 score of the primitives and extrusions of the predicted CAD sequence with the ground truth sequence; 4) calculating the Invalid Ratio (IR) to evaluate the proportion of generated invalid CAD

| User Level | Method | GPT-4V↑ | CD↓ | | JSD↓ | MMD↓ | COV↑ | F1↑ | | IR↓ |
| --- | --- | --- | --- | --- | --- | --- | --- | --- | --- | --- |
| | | | Mean | Median | | | | Sketch | Extrusion | |
| Abstract (L0) | GPT-4o | 7.60 | 338.52 | 247.11 | 207.71 | 40.78 | 10.37 | 16.12 | 63.51 | 70.35 |
| | LLaMA3.1 | 10.70 | 262.26 | 201.16 | 176.36 | 35.85 | 19.32 | 22.54 | 58.61 | 33.46 |
| | DeepCAD | 13.50 | 253.06 | 191.44 | 137.89 | 30.10 | 18.77 | 30.49 | 67.07 | 14.96 |
| | Text2CAD | 23.90 | 233.84 | 187.31 | 130.13 | 24.06 | 24.22 | 39.50 | 85.30 | 1.78 |
| | **PTA (Ours)** | **44.30** | **183.14** | **113.79** | **26.35** | **19.69** | **40.63** | **50.61** | **90.33** | **0.63** |
| Beginner (L1) | GPT-4o | 6.80 | 278.75 | 232.70 | 187.62 | 39.36 | 9.66 | 16.76 | 72.34 | 73.71 |
| | LLaMA3.1 | 13.20 | 269.45 | 201.41 | 156.35 | 40.11 | 17.37 | 23.63 | 65.34 | 43.52 |
| | DeepCAD | 18.60 | 281.35 | 232.33 | 139.00 | 32.65 | 15.89 | 32.77 | 76.06 | 16.61 |
| | Text2CAD | 20.50 | 244.84 | 206.28 | 129.40 | 22.91 | 21.87 | 43.74 | 91.75 | 1.50 |
| | **PTA (Ours)** | **40.90** | **200.25** | **142.50** | **25.52** | **18.66** | **37.50** | **52.07** | **94.53** | **0.68** |
| Intermedia. (L2) | GPT-4o | 8.00 | 260.63 | 200.36 | 103.66 | 32.44 | 20.79 | 22.10 | 73.23 | 81.47 |
| | LLaMA3.1 | 18.40 | 214.70 | 115.31 | 87.99 | 30.51 | 25.66 | 29.38 | 72.29 | 37.99 |
| | DeepCAD | 15.30 | 210.79 | 111.06 | 50.34 | 25.36 | 27.21 | 40.76 | 62.51 | 15.63 |
| | Text2CAD | 24.60 | 150.04 | 74.16 | 30.65 | 17.33 | 40.65 | 48.55 | 93.40 | 1.77 |
| | **PTA (Ours)** | **33.70** | **140.53** | **67.73** | **11.91** | **16.95** | **43.75** | **52.55** | **94.20** | **1.37** |
| Expert (L3) | GPT-4o | 11.20 | 83.52 | 44.83 | 63.99 | 13.42 | 40.83 | 32.07 | 73.86 | 66.91 |
| | LLaMA3.1 | 14.90 | 75.91 | 20.24 | 30.03 | 8.57 | 44.36 | 43.69 | 67.04 | 26.57 |
| | DeepCAD | 17.60 | 69.34 | 8.24 | 20.45 | 7.66 | 53.21 | 62.67 | 54.86 | 12.81 |
| | Text2CAD | 25.80 | 26.41 | 0.37 | 3.15 | 5.18 | 73.34 | 63.80 | 93.31 | 0.93 |
| | **PTA (Ours)** | **30.50** | **20.08** | **0.31** | **2.85** | **4.76** | **78.12** | **64.95** | **94.59** | **0.71** |

Table 1: Comparisons with state-of-the-art command sequence methods, open-source LLM, and closed-source LLM. Mean CD, Median CD, JSD, and MMD are multiplied by $10^3$. COV is multiplied by $10^2$. GPT-4V Evaluation, F1, and IR are multiplied by 100%.

sequences. Additionally, following (Xu et al., 2022; 2023), we also use 5) the Jensen-Shannon Divergence (JSD), which measures the distribution similarity between the generated CAD model and the Ground Truth. 6) Coverage (COV), which quantifies the percentage of real data covered by generated samples; 7) Minimum Matching Distance (MMD), which evaluates the closest match between generated samples and real data.

### 5.1.4 COMPARED METHODS

We compare PTA with the available SOTA Text-to-CAD method Text2CAD (Khan et al., 2024b), the classic VAE-based CAD sequence generation method DeepCAD (Wu et al., 2021), the open-source LLM LLaMA3.1-8B (Dubey et al., 2024), and the closed-source LLM GPT-4o (Hurst et al., 2024). To make DeepCAD suitable for text input, we replace its CAD encoder with a pre-trained BERT. To ensure a fair comparison, except for GPT-4o, we train or fine-tune all methods on the Text2CAD trainset. And we evaluate all methods on the Text2CAD test set. Although there are other Text-to-CAD methods (Wang et al., 2025b; Li et al., 2025a; 2024b; Liao et al., 2025), their official implementation code is not open-sourced, we can not conduct a fair comparison with them.

## 5.2 MAIN RESULTS

### 5.2.1 QUALITATIVE EVALUATION

We present the visual results generated by our PTA and the comparison methods, as shown in Figure 4, We observe that GPT-4o, LLaMA3.1, and DeepCAD can only perceive simple general shapes, exhibiting poor generation results. Text2CAD may capture some details, but it still struggles to accurately generate local details and complex shapes ( as shown in rows 2,3,4 of Figure 4). In contrast, our PTA can faithfully follow the user instructions, achieving results that are closest to Ground Truth in terms of both overall shapes and local details. If the user instructions provide detailed operation steps and parameters (cf. the last row of Figure 4), both our PTA and Text2CAD can produce results close to the ground truth (GT). However, in practice, this level of user instruction imposes significant input pressure on users and is not user-friendly for non-professionals. Additional visual results are provided in the Appendix A.7.

| User Level | Method | CD↓ | | Avg. F1↑ | IR↓ |
|---|---|---|---|---|---|
| | | Mean | Median | | |
| Abstract (L0) | Planner only | 248.93 | 218.97 | 41.87 | 20.06 |
| | Actioner only | 247.21 | 200.32 | 45.16 | 2.66 |
| | **PTA (Ours)** | **183.14** | **113.79** | **60.54** | **0.63** |
| Beginner (L1) | Planner only | 247.33 | 206.71 | 46.66 | 25.37 |
| | Actioner only | 266.57 | 189.59 | 53.34 | 4.71 |
| | **PTA (Ours)** | **200.25** | **142.50** | **62.68** | **0.68** |
| Intermedia. (L2) | Planner only | 199.36 | 93.38 | 53.73 | 18.90 |
| | Actioner only | 180.72 | 79.74 | 55.48 | 2.17 |
| | **PTA (Ours)** | **140.53** | **67.73** | **62.96** | **1.37** |
| Expert (L3) | Planner only | 45.78 | 15.33 | 52.91 | 15.66 |
| | Actioner only | 37.13 | 0.45 | 67.89 | 1.14 |
| | **PTA (Ours)** | **20.08** | **0.31** | **72.36** | **0.71** |

Table 2: Comparison between single-stage generation and our bi-level generation (PTA)

| User Level | Guidance in Actioner | CD↓ | | Avg. F1↑ | IR↓ |
|---|---|---|---|---|---|
| | | Mean | Median | | |
| Abstract (L0) | $F_{plan}$ only | 280.07 | 259.31 | 28.82 | 3.05 |
| | Concat | 198.36 | 159.98 | 55.91 | 2.32 |
| | Add | 208.06 | 163.27 | 55.17 | 2.92 |
| | **RAM** | **183.14** | **113.79** | **60.54** | **0.63** |
| Beginner (L1) | $F_{plan}$ only | 299.62 | 238.04 | 40.63 | 3.57 |
| | Concat | 228.72 | 171.87 | 56.99 | 1.78 |
| | Add | 227.51 | 180.91 | 57.11 | 1.24 |
| | **RAM** | **200.25** | **142.50** | **62.68** | **0.68** |
| Intermediate (L2) | $F_{plan}$ only | 259.79 | 216.74 | 47.43 | 3.66 |
| | Concat | 147.66 | 70.16 | 60.14 | 1.97 |
| | Add | 151.02 | 69.77 | 59.90 | 0.93 |
| | **RAM** | **140.53** | **67.73** | **62.96** | **1.37** |
| Expert (L3) | $F_{plan}$ only | 200.33 | 179.31 | 59.82 | 2.31 |
| | Concat | 24.38 | 0.36 | 71.44 | 0.93 |
| | Add | 24.84 | 0.35 | 71.62 | 1.01 |
| | **RAM** | **20.08** | **0.31** | **72.36** | **0.71** |

Table 3: Ablation Study of RAM.

### 5.2.2 QUANTITATIVE EVALUATION

Based on the Text2CAD dataset (Khan et al., 2024b), our experiments are conducted across 4 levels of User Instruction. As shown in Table 1, our PTA achieves higher GPT-4V evaluation scores across all user levels compared to the baseline methods, demonstrating that the CAD models generated by PTA more faithfully match user instructions. Additionally, PTA achieves the lowest CD, JSD, MMD, and the highest COV, F1 score. Specifically, at the Abstract and Beginner levels, Median CD improved by 39% (187.31→113.79) and 31% (206.28→142.50), JSD improved by 79%(130.13→26.35) and 80%(129.40→25.52) while the Sketch F1 scores improved by 11.11 (39.50→50.61) and 8.33 (43.74→52.07), respectively. For the Intermediate and Expert levels, user instructions already provide some operational processes (in practice, these level of user instruction increases the user's input burden), which may overlap to some extent with the high-level plan. However, our PTA can better utilize planning sequences and user-provided requirements, resulting in performance improvements. It is worth noting that PTA achieves an extremely low invalid ratio, with an average IR of 0.85% across the four user levels. In other words, the success rate of the CAD sequences generated by PTA reaches an impressive 99.15%. We observe that LLMs (GPT-4o, LLaMA3.1) are not adept at outputting task-specific low-level CAD command sequences, resulting in high invalidity rates.

### 5.3 ANALYSIS AND DISCUSSIONS

We conduct ablation studies to analyze our PTA. We first compare bi-level generation with single-stage generation. And then we explore the efficiency of the designed RAM. Details and visual results of ablation experiments, more analysis, are provided in the Appendix A.2 A.4 A.5 A.6.

### 5.3.1 DISCUSSION ABOUT BI-LEVEL GENERATION

We conduct ablation experiments to validate the superiority of bi-level generation. The experimental results are shown in Table 2. We find that when fine-tuning the Planner model (Qwen3-8B) to directly generate low-level command sequences based on user instructions(Planner Only), it produces results similar to other LLMs (LLaMA3.1-8B, GPT-4o), exhibiting a high Invalid Ratio. When we remove the high-level plan input from Actioner, demanding it to use only user instructions to generate command sequences (Actioner Only), it performs significantly worse than the complete PTA in metrics such as CD and F1. These results demonstrate the efficacy of bi-level generation.

### 5.3.2 DISCUSSION ABOUT RAM

We explore how to efficiently utilize the requirement information from user instructions and the operation information from high-level plans to guide low-level command sequence generation. As shown in Table 3, if using only high-level plan features as guidance to generate command sequences ($F_{plan}$ only), due to the absence of requirement information, it struggles to generate CAD sequences

| Plan type | CD↓ | | F1↑ | | IR↓ |
|---|---|---|---|---|---|
| | Mean | Median | Sketch | Extrusion | |
| Nli_data | 220.61 | 156.00 | 39.88 | 84.85 | 2.46 |
| High-level plan | **183.14** | **113.79** | **50.61** | **90.33** | **0.63** |

Table 4: Comparison between nli_data plan and high-level operation plan.

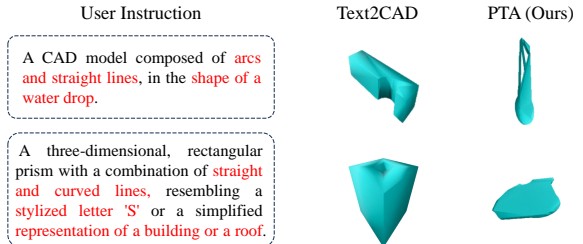

Figure 5: Failure cases. Red text indicates information that requires special attention.

that satisfy the user requirements. Especially, since the user instructions of L2,3 provide more information, their performance worsens significantly after the user instruction input is removed. Additionally, compared to concatenating (Concat) or adding (Add) plan features and instruction features, the guidance vector obtained through our requirement-aware mechanism (RAM) can fully utilize requirement information in the user instructions, achieving the lowest CD and highest F1.

### 5.3.3 DISCUSSION ABOUT HIGH-LEVEL PLAN

In this section, we provide an analysis of the high-level plans and the nli_data in the dataset. The nli_data serves as a natural language translation of the command sequence, containing low-level control actions and specific parameters. In contrast, the high-level plan represents a complete operational flow, where each operation step generalizes the underlying control actions without including detailed parameters (cf. Appendix Figure 7). We analyze whether nli_data can be used as plan. We retrain PTA using nli_data in place of the high-level plan, and the results are shown in Table 4. We find that using nli_data yields inferior results compared to the high-level plan. The reason is as follows: when using nli_data as plan, the Actioner merely acts as a translator from natural language to command sequence, while the LLM-based Planner must generate a complete and detailed low-level operational process with accurate parameters from the user instruction. Due to the inherent hallucination issues in LLMs, this process is highly challenging, and the generated nli_data plan often contains erroneous information, resulting in inaccurate command sequences. In contrast, with the high-level plan, the Planner only needs to extract a global operational flow, while the Actioner perceives requirement information from the user instruction to supplement operation details and parameters. This approach enables PTA to achieve a more comprehensive and accurate parsing of information, leading to improved performance.

### 5.3.4 ANALYSIS OF FAILURE CASES

As shown in Figure 5, we observe that when user instructions contain descriptions that are uncommon in the training set (*e.g.*, "shape of a water drop", "representation of a building or a roof"), PTA struggles to generate CAD models that fully meet user requirements. However, thanks to our LLM-based Planner and the introduced requirement-aware mechanism, PTA still perceives some basic shapes and requirements (*e.g.*, composed of arcs and straight lines, a combination of straight and curved lines). A simple solution to address this issue is to perform data augmentation for rare descriptions during training.

## 6 CONCLUSION

In this paper, we introduce PTA, a bi-level CAD command sequence generation method. We employ a Planner to generate a high-level operation plan based on user instruction, and then develop an Actioner that maps the high-level plan to a low-level executable command sequence. Experiment validates the efficiency of PTA. In the future, we aim to explore the potential of PTA in Multimodality-Conditioned CAD Generation.

## 7 ETHICS STATEMENT

This research aims to the automatic CAD models generation to assist professional design in the CAD domain. The data we used is developed using LLMs based on existing publicly available datasets. This process does not involve human participation, and the developed data is solely for research purposes in CAD generation technology.

## 8 REPODUCIBILITY STATEMENT

We have made every effort to ensure the reproducibility of our research. Our PTA is detailed in Section (Section 4). All implementation details, including model architecture and training hyper parameters are provided in the Experimental Setting (Section 5.1). The data we used is developed based on existing publicly available datasets, with details provided in the Appendix A.2. To facilitate direct replication and further research, we will release our source code, the used dataset, and model checkpoints upon publication, contributing to the open-source community.

## 9 ACKNOWLEDGEMENTS

This work was supported by National Natural Science Foundation of China (No.62576139, 62176093), National Key Research and Development Program of China (No.2023YFC3502900).

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

# A  APPENDIX

## A.1  CAD COMMAND SEQUENCE

Following Wu et al. (2021); Wang et al. (2025b); Khan et al. (2024a;b), the CAD model is represented as a sequence of modeling operations executed by the user to construct a 3D shape. The sequence stores key modeling commands and parameters (cf. Figure 6) in the order of CAD construction. This representation uses a sketch-and-extrude format, where each 2D sketch is represented as a closed loop composed of multiple curves. Following Khan et al. (2024a;b), we represent sketches using a sequence of 2D coordinates, adding endpoint markers at curves, loops, faces, and the end of sketches. For extrusion commands, each command is marked by 11 parameters.

| Commands | Parameters |
|---|---|
| <SOL> | $\emptyset$ |
| L (Line) | $x, y$ : line end-point |
| A (Arc) | $x, y$ : arc end-point
$\alpha$ : sweep angle
$f$ : counter-clockwise flag |
| R (Circle) | $x, y$ : center
$r$ : radius |
| E (Extrude) | $\theta, \phi, \gamma$ : sketch plane orientation
$p_x, p_y, p_z$ : sketch origin
$s$ : scale of associated sketch profile
$e_1, e_2$ : extrude distances toward both sides
$b$ : Boolean type,    $u$ : extrude type |
| <EOS> | $\emptyset$ |

Figure 6: CAD commands and their main parameters. <SOL>indicates the start of a loop; <EOS>indicates the end of the whole sequence.

## A.2  HIGH-LEVEL PLAN

### A.2.1  CONSTRUCTION OF HIGH-LEVEL PLAN

In our bi-level generation, we need to construct data triplets of (user instruction, high-level plan, command sequence). Our experiments are conducted on the Text2CAD dataset (Khan et al., 2024b). In the Text2CAD dataset, in addition to user instructions and command sequences, it provides a detailed text-based action sequence and operation parameters for each command sequence, named as nli_data. As shown in Figure 7, we provide a prompt and high-level plan example to Qwen2.5-32B-instruct (Qwen et al., 2025), guiding it to extract or aggregate high-level operation plan from the nli_data. We construct high-level plan data for the Text2CAD datasets and use it as real high-level plan data to train our PTA.

### A.2.2  DISCUSSION ABOUT THE QUALITY OF HIGH-LEVEL PLAN

In this section, we evaluate the quality of the constructed high-level plans. First, inspired by Chen et al. (2025c); Gong et al. (2025), we conduct a user study. In addition, we perform comparative experiments with OpenAI o3. For the user study, we randomly select 1,000 high-level plan samples, which are evenly divided into five groups. We then invite 30 volunteers with at least a bachelor's degree, and every volunteer is assigned to one data group. For each sample, participants are presented with the user instruction and nli_data, and then asked to judge whether the corresponding high-level plan is satisfactory (binary classification: any unreasonable aspect is counted as unsatisfactory). In total, we collect 6,000 responses, with a satisfaction rate of 98.75%.

For the comparative experiments, we employ an advanced closed-source LLM OpenAI o3 (OpenAI, 2025) to generate high-level plans and resulting 100,000 triplets of (user instruction, high-level plan,

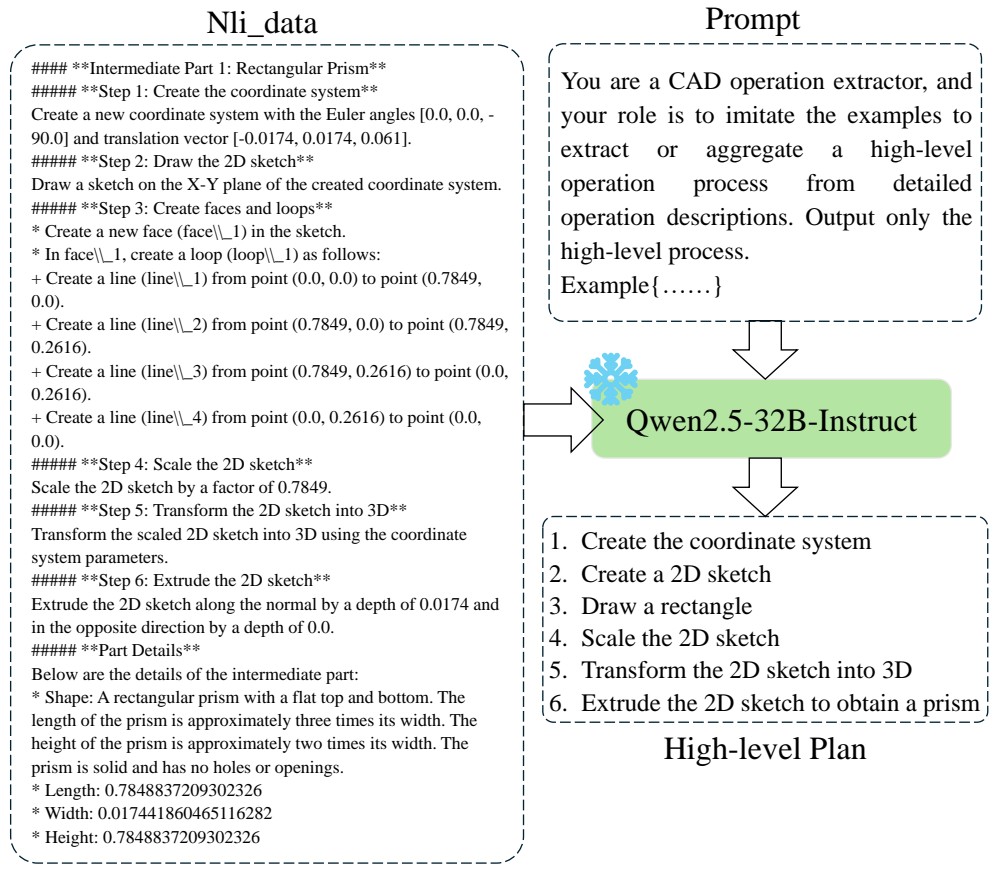

Figure 7: Construction process of our high-level operation plan.

command sequence). Using this dataset, we retrain our PTA. Similarly, we also retrain PTA using the 100,000 triplets constructed with Qwen2.5-32B-Instruct. The experimental results are provided in the Table 8. The results show that they achieve comparable performance. Considering that plan generation involves constructing a complete high-level workflow from detailed operational steps, which is not a difficult task, and Qwen2.5-32B-instruct is capable of accomplishing it.

In summary, we believe that the high-level plans generated by Qwen2.5-32B-Instruct are of high quality and can be used to train our method.

### A.2.3 IMPACT OF FINE-TUNING ON THE PLANNER

We present the performance changes of the Planner before and after fine-tuning. As shown in Table 6, we prompt base Planner model (Qwen3-8B) without fine-tuning to generate high-level plans based on user instructions (Base Planner) and use these plans to guide the generation of command sequences. The results demonstrate that after fine-tuning, the performance has significantly improved. The results demonstrate that, compared to using only prompts (Base Planner), the fine-tuned Planner (Prof. Planner) achieves better performance, validating the effectiveness of fine-tuning.

### A.2.4 DISCUSSION ON THE BACKBONE LLM FOR THE PLANNER

We also compare the results of using Qwen2.5-7B-Instruct and Qwen3-8B as the base LLM for the Planner, where both are fine-tuned using high-level plan. The experimental results are shown in Table 5. The results indicate that Qwen3-8B performed slightly better than Qwen2.5-7B-Instruct, though the difference was not significant.

| User Level | Base LLM | CD↓ | | Avg. | IR↓ |
| | | Mean | Median | F1↑ | |
|---|---|---|---|---|---|
| Abstract | Qwen2.5-7B-Instruct | 191.21 | 115.97 | 58.92 | 1.13 |
| (L0) | Qwen3-8B | **183.14** | **113.79** | **60.54** | **0.63** |

Table 5: Comparison of the Planner Foundations.

| User Level | Plan | CD↓ | | Avg. | IR↓ |
| | | Mean | Median | F1↑ | |
|---|---|---|---|---|---|
| Abstract | Base Planner | 190.02 | 124.51 | 52.00 | 1.62 |
| (L0) | **Prof. Planner** | **183.14** | **113.79** | **60.54** | **0.63** |
| Beginner | Base Planner | 223.43 | 158.47 | 50.32 | 1.41 |
| (L1) | **Prof. Planner** | **200.25** | **142.50** | **62.68** | **0.68** |
| Intermediate | Base Planner | 181.68 | 79.61 | 50.80 | 2.73 |
| (L2) | **Prof. Planner** | **140.53** | **67.73** | **62.96** | **1.37** |
| Expert | Base Planner | 44.81 | 0.45 | 68.07 | 1.88 |
| (L3) | **Prof. Planner** | **20.08** | **0.31** | **72.36** | **0.71** |

Table 6: Comparison of the Planner Before and After Fine-Tuning.

## A.3 COMPARED METHODS

We provide implementation details of compared methods, including Text2CAD (Khan et al., 2024b), DeepCAD (Wu et al., 2021), the open-source LLM LLaMA3.1-8B (Dubey et al., 2024), and the closed-source large model GPT-4o (Hurst et al., 2024). While there are other Text-to-CAD methods such as CAD-GPT (Wang et al., 2025b), CAD-MLLM (Xu et al., 2024), TCAD-Gen (Liao et al., 2025), their official code are currently not available, and we don't compare them with PTA.

**Text2CAD.** Text2CAD is the first method to generate CAD sequences based on textual input. We use official code for reproduction and conduct a fair comparison with PTA.

**DeepCAD.** DeepCAD first represents CAD models as a sequence of sketch primitives and 3D extrusion operations. It employs a transformer encoder to encode CAD models into high-dimensional vectors and a decoder to reconstruct CAD models. To adapt DeepCAD for textual input, we replace the original CAD encoder with a pre-trained BERT (Devlin et al., 2019), while retaining its original decoder to generate CAD command sequences. Other implementation details remain consistent with the official code.

**LLaMA3.1-8B.** We perform full-parameter fine-tuning for LLaMA3.1-8B (Dubey et al., 2024) using the User Instruction-Command sequence pairs. It is worth noting that Transformer-based CAD generation methods (Wu et al., 2021; Khan et al., 2024a) typically tokenize the command sequence, whereas LLM-based generation methods (Wang et al., 2025b; Xu et al., 2024) output the command sequence in text JSON. Therefore, for LLaMA3.1, we use the text JSON for the command sequence.

**GPT-4o.** We provide a prompt along with a user instruction-command sequence example to guide GPT-4o in generating the command sequence based on the given user instruction. Consistent with LLaMA3.1-8B, the command sequence is formatted in text JSON.

## A.4 MORE COMPARISONS WITH LLMS

In this section, we provide implementation details of the experiments on LLM-based command sequence generation. For the closed-source LLMs GPT-4o, OpenAI o3 (OpenAI, 2025), OpenAI o4-mini (OpenAI, 2025), GPT-5 (Singh et al., 2025), Gemini-2.5-pro, we provide a prompt with an example (cf. Figure 9), asking it to generate the corresponding Command sequence based on the given user instruction. For LLaMA3.1-8B (Dubey et al., 2024) and Qwen3-8B (Yang et al., 2025a), we use

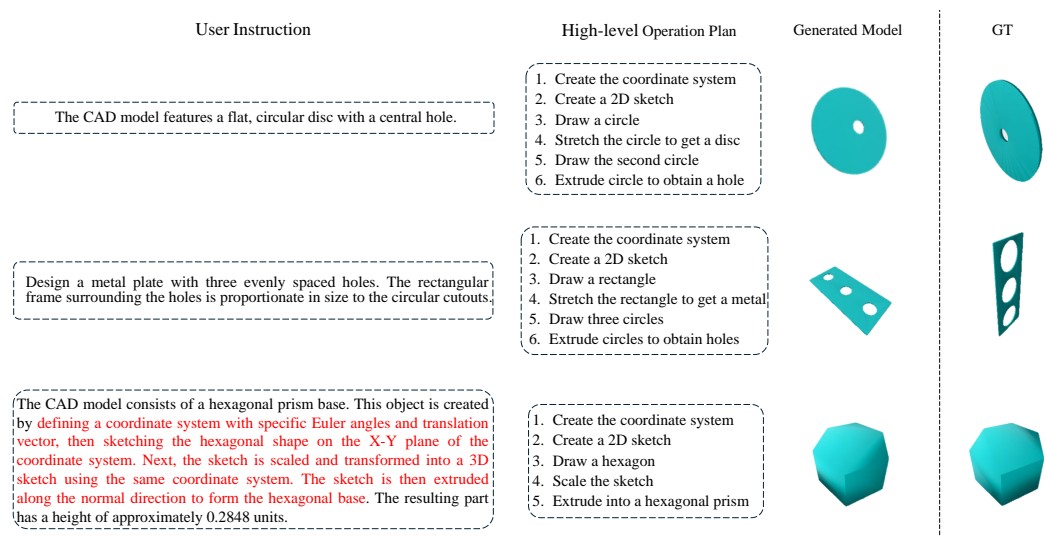

Figure 8: Illustration of user instructions and high-level plans. The third one is an intermediate-level sample, and red text indicates operation processes provided in its user instructions.

> **Prompt:** You are a CAD command sequence generator, and you need to generate CAD command sequences based on user instructions. The command sequence involves lines, arcs, and circles, each marked by the endpoints, and extrusions. There is an example:
>   example: {......}
>    Now, transform the user instruction into command sequence using the provided example as reference. Output only the command sequence data. Your output does not require extra line breaks, quotation marks, or any prefixes.

Figure 9: An example prompt is provided for the LLMs.

the same prompt and perform full-parameter supervised fine-tuning (3 epochs, AdamW optimizer with learning rate $10^{-4}$). Additionally, we apply direct preference optimization (DPO) (Rafailov et al., 2023) to the finetuned Qwen3-8B; specifically, we render multiple CAD sequences generated by the finetuned Qwen3-8B into images, and Qwen3-VL-8b-Instruct (Bai et al., 2025) scores the CAD images according to the user instructions to construct 10000 preference pairs.

The experimental results are shown in Table 7. All closed-source LLMs resulted in extremely high invalid rates (IR). For the fine-tuned LLaMA3.1-8B and Qwen3-8B, their IRs show some reduction but remain relatively high. Furthermore, when finetuned Qwen3-8B is further optimized using DPO (Qwen3*), the IR only exhibited a slight decrease. These results demonstrate that directly generating command sequences with LLMs remains challenging.

## A.5    IMPLEMENTATIONS OF ABLATION STUDIES

### A.5.1    ABLATION STUDY OF BI-LEVEL GENERATION

We conduct ablation experiments on PTA to validate the effectiveness of bi-level generation. The complete PTA includes the Planner and the Actioner. The Planner parses user instructions to generate high-level operation plans, while the Actioner generates low-level executable command sequences. In the ablation experiments, we discard the bi-level generation and request either the Planner or the Actioner to directly generate command sequences based on user instructions. We provide detailed implementations. Visual results are provided in Figure 10.

**Planner only.** We use the Planner base model (Qwen3-8B) and perform full-parameter fine-tuning (SFT) using User Instruction-Command Sequence data, requesting it to generate low-level executable sequences based on user instructions. Following LLM-based method (Wang et al., 2025b), instead of tokenizing the command sequences, we retain the text JSON outputs.

| User Level | Method | CD↓ | | JSD↓ | MMD↓ | COV↑ | F1↑ | | IR↓ |
|---|---|---|---|---|---|---|---|---|---|
| | | Mean | Median | | | | Sketch | Extrusion | |
| Abstract (L0) | GPT-4o | 338.52 | 247.11 | 207.71 | 40.78 | 10.37 | 16.12 | 63.51 | 70.35 |
| | OpenAI o3 | 312.99 | 163.39 | 250.35 | 33.93 | 22.22 | 26.17 | 61.65 | 68.09 |
| | OpenAI o4-mini | 284.12 | 218.36 | 292.87 | 27.60 | 15.00 | 25.67 | 55.78 | 77.25 |
| | GPT-5 | 269.73 | 197.63 | 295.39 | 25.86 | 27.35 | 37.06 | 64.84 | 67.99 |
| | Gemini-2.5-pro | 237.78 | 186.79 | 286.59 | 24.68 | 26.31 | 35.75 | 66.31 | 85.77 |
| | LLaMA3.1 | 262.26 | 201.16 | 176.36 | 35.85 | 19.32 | 22.54 | 58.61 | 33.46 |
| | Qwen3 | 248.93 | 218.97 | 179.87 | 28.94 | 21.33 | 31.92 | 71.73 | 20.06 |
| | Qwen3* | 226.94 | 193.07 | 167.86 | 26.29 | 22.47 | 33.86 | 76.20 | 18.29 |
| | **PTA (Ours)** | **183.14** | **113.79** | **26.35** | **19.69** | **40.63** | **50.61** | **90.33** | **0.63** |
| Beginner (L1) | GPT-4o | 278.75 | 232.70 | 187.62 | 39.36 | 9.66 | 16.76 | 72.34 | 73.71 |
| | OpenAI o3 | 246.92 | 170.08 | 144.95 | 49.07 | 10.25 | 16.10 | 60.26 | 78.27 |
| | OpenAI o4-mini | 287.29 | 207.93 | 192.71 | 33.95 | 10.71 | 16.56 | 61.12 | 86.51 |
| | GPT-5 | 226.82 | 185.92 | 142.52 | 36.37 | 14.70 | 19.53 | 62.05 | 70.22 |
| | Gemini-2.5-pro | 252.79 | 171.92 | 154.59 | 34.19 | 13.11 | 13.96 | 57.99 | 70.92 |
| | LLaMA3.1 | 269.45 | 201.41 | 156.35 | 40.11 | 17.37 | 23.63 | 65.34 | 43.52 |
| | Qwen3 | 247.33 | 206.71 | 195.09 | 40.09 | 22.47 | 38.48 | 71.20 | 25.37 |
| | Qwen3* | 206.62 | 185.68 | 157.66 | 38.78 | 27.29 | 41.68 | 76.64 | 24.05 |
| | **PTA (Ours)** | **200.25** | **142.50** | **25.52** | **18.66** | **37.50** | **52.07** | **94.53** | **0.68** |
| Intermedia. (L2) | GPT-4o | 260.63 | 200.36 | 103.66 | 32.44 | 20.79 | 22.10 | 73.23 | 81.47 |
| | OpenAI o3 | 208.85 | 159.86 | 107.69 | 37.02 | 23.33 | 29.41 | 73.69 | 68.33 |
| | OpenAI o4-mini | 203.38 | 184.75 | 114.08 | 40.67 | 25.44 | 29.80 | 72.91 | 65.29 |
| | GPT-5 | 183.22 | 166.09 | 102.65 | 35.24 | 20.00 | 37.67 | 69.86 | 65.89 |
| | Gemini-2.5-pro | 195.99 | 170.28 | 112.92 | 29.33 | 19.51 | 31.77 | 71.35 | 62.81 |
| | LLaMA3.1 | 214.70 | 115.31 | 87.99 | 30.51 | 25.66 | 29.38 | 72.29 | 37.99 |
| | Qwen3 | 199.36 | 93.38 | 74.03 | 26.29 | 31.15 | 45.38 | 78.79 | 18.90 |
| | Qwen3* | 187.88 | 92.09 | 70.54 | 26.23 | 35.66 | 50.11 | 80.12 | 14.05 |
| | **PTA (Ours)** | **140.53** | **67.73** | **11.91** | **16.95** | **43.75** | **52.55** | **94.20** | **1.37** |
| Expert (L3) | GPT-4o | 83.52 | 44.83 | 63.99 | 13.42 | 40.83 | 32.07 | 73.86 | 66.91 |
| | OpenAI o3 | 86.34 | 38.86 | 65.93 | 12.49 | 46.67 | 44.37 | 62.84 | 65.32 |
| | OpenAI o4-mini | 72.09 | 54.40 | 78.91 | 12.27 | 45.00 | 44.55 | 72.42 | 77.82 |
| | GPT-5 | 60.47 | 40.81 | 79.26 | 9.75 | 58.63 | 54.30 | 62.51 | 75.04 |
| | Gemini-2.5-pro | 68.85 | 35.83 | 75.29 | 10.85 | 55.33 | 54.77 | 69.52 | 82.93 |
| | LLaMA3.1 | 75.91 | 20.24 | 30.03 | 8.57 | 44.36 | 43.69 | 67.04 | 26.57 |
| | Qwen3 | 45.78 | 15.33 | 28.67 | 8.09 | 45.45 | 46.39 | 72.47 | 15.66 |
| | Qwen3* | 37.65 | 14.65 | 27.04 | 6.23 | 52.98 | 50.11 | 77.23 | 14.05 |
| | **PTA (Ours)** | **20.08** | **0.31** | **2.85** | **4.76** | **78.12** | **64.95** | **94.59** | **0.71** |

Table 7: Comparisons with LLMs.

**Actioner only.** After discarding the high-level plan generation stage, the high-level operation plan is unavailable. Therefore, we only input user instructions into Actioner. The instruction features are used to guide the Transformer decoder in generating the command sequence.

### A.5.2 ABLATION STUDY OF RAM

To analyze the requirement-aware mechanism (RAM), we use guidance obtained through different strategies to direct the Transformer decoder in generating the command sequence. The implementation details of the ablated versions are provided below. We present visual results in Figure 11.

$F_{plan}$ **only.** In this ablated version, we explore whether relying only on the high-level plan can accurately generate low-level executable command sequences. We discard the original user instruction input for the Actioner and use only the high-level plan features to guide Transformer decoder. Experimental results (cf. Table 3 of main paper) demonstrate that without the requirement information provided in user instructions, it becomes challenging to accurately generate the command sequences.

**Concat.** We retain both the user instruction and high-level plan inputs. After obtaining the instruction features $F_{inst}$ and plan features $F_{plan}$, we concatenate them along the length and use the

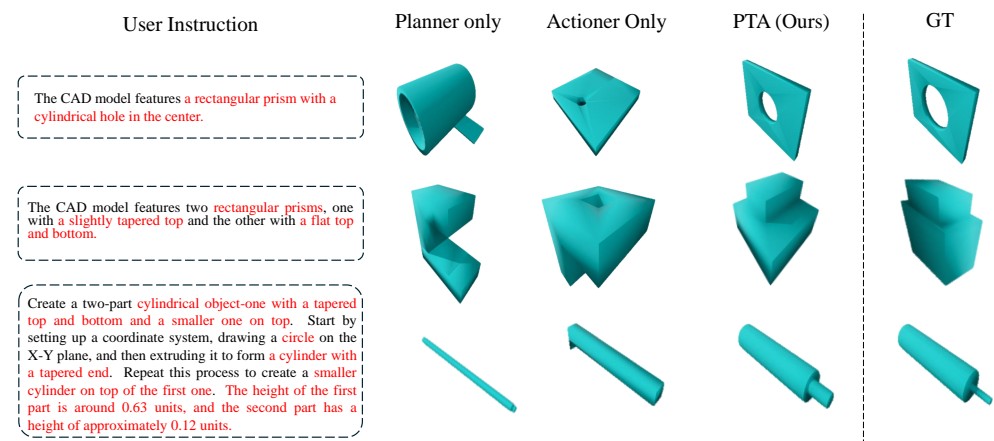

Figure 10: Visual ablation results of bi-level generation. Red text indicates information that requires special attention.

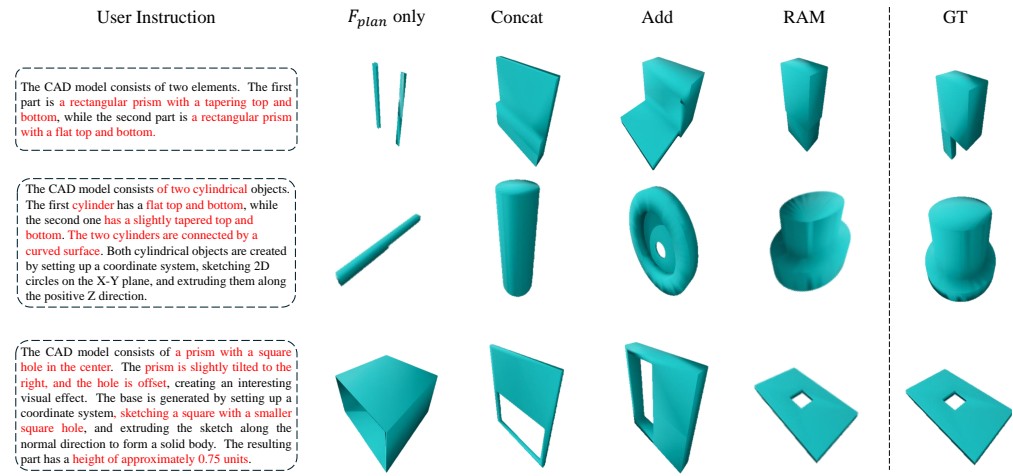

Figure 11: Visual ablation results of our introduced RAM. Red text indicates information that requires special attention.

concatenated vector to guide the generation of low-level command sequences. Experimental results show that (cf. Table 3 of main paper), compared to the "$F_{plan}$ only", incorporating constraint information in the user instructions significantly improves performance. However, it still falls short of the results achieved by our RAM.

**Add.** In this version, we still retain both the user instruction and high-level plan inputs. We add the instruction features $F_{inst}$ and plan features $F_{plan}$ together as the guidance information for the Transformer decoder, which demonstrates results similar to "Concat".

## A.6 DISCUSSION ABOUT INCORPORATING REASONING STEPS

### A.6.1 IMPACT OF INTERMEDIATE REASONING STEPS

To investigate the impact of reasoning steps, we conduct experiments to compare the difference between plans with and without the incorporation of reasoning steps. Specifically, we use Qwen2.5-32B-Instruct and OpenAI o3 to separately construct plans with reasoning steps (called reasoning plans, cf. Figure 12), each resulting in 100,000 triplets of (user instruction, reasoning plan, command sequence). These datasets are then used to train PTA, respectively. Similarly, we train PTA using the

| User Level | Plan Type | Source | CD↓ | | JSD↓ | MMD↓ | COV↑ | F1↑ | | IR↓ |
|---|---|---|---|---|---|---|---|---|---|---|
| | | | Mean | Median | | | | Sketch | Extrusion | |
| Abstract (L0) | Reasoning | Qwen2.5-32B | 207.01 | 125.03 | 43.60 | 23.83 | 36.03 | 36.33 | 86.35 | 2.61 |
| | | OpenAI o3 | 208.25 | 127.65 | 44.74 | 22.79 | 33.59 | 35.77 | 87.24 | 2.18 |
| | High-level | Qwen2.5-32B | 206.09 | 126.06 | 43.55 | 22.39 | 37.15 | 36.92 | 87.42 | 2.53 |
| | | OpenAI o3 | 207.75 | 126.45 | 43.13 | 23.90 | 35.15 | 35.28 | 88.21 | 2.80 |
| Beginner (L1) | Reasoning | Qwen2.5-32B | 235.89 | 181.89 | 33.03 | 20.74 | 26.90 | 41.33 | 89.66 | 3.65 |
| | | OpenAI o3 | 228.62 | 170.15 | 35.98 | 26.99 | 28.12 | 41.77 | 90.81 | 2.75 |
| | High-level | Qwen2.5-32B | 231.60 | 175.66 | 31.76 | 23.46 | 25.00 | 41.14 | 92.93 | 3.47 |
| | | OpenAI o3 | 237.94 | 173.79 | 32.74 | 20.17 | 25.15 | 39.24 | 90.15 | 3.52 |
| Intermedia. (L2) | Reasoning | Qwen2.5-32B | 160.14 | 78.40 | 16.26 | 18.47 | 32.28 | 50.47 | 90.11 | 1.66 |
| | | OpenAI o3 | 158.59 | 77.88 | 16.73 | 16.99 | 31.40 | 53.23 | 92.42 | 1.04 |
| | High-level | Qwen2.5-32B | 154.02 | 74.06 | 14.99 | 18.86 | 37.81 | 51.03 | 92.31 | 1.17 |
| | | OpenAI o3 | 155.11 | 76.22 | 12.27 | 19.68 | 34.53 | 50.12 | 92.60 | 1.10 |
| Expert (L3) | Reasoning | Qwen2.5-32B | 38.68 | 0.45 | 4.12 | 6.57 | 64.84 | 58.44 | 92.14 | 3.05 |
| | | OpenAI o3 | 34.66 | 0.59 | 4.62 | 5.79 | 66.40 | 57.65 | 92.57 | 2.08 |
| | High-level | Qwen2.5-32B | 33.81 | 0.48 | 3.77 | 5.26 | 69.68 | 62.27 | 93.67 | 1.26 |
| | | OpenAI o3 | 33.23 | 0.51 | 3.89 | 5.10 | 72.21 | 61.55 | 91.99 | 2.46 |

Table 8: Discussion about incorporating reasoning steps.

datasets of 100,000 high-level plans without reasoning steps. The results are shown in Table 8. The performance achieved by the reasoning plan is comparable to that of the high-level plan. In other words, integrating reasoning steps into the plan has minimal impact on performance.

The difference between the reasoning plan and the high-level plan is that the reasoning plan includes reasoning steps, explicitly expressing the operational intent and the logic between operations. As for our PTA, there is a requirement-aware mechanism (RAM) present during the command sequence generation. In the RAM, cross-attention integrates requirement information from the user instruction to obtain operational intent, while self-attention captures contextual relationships to understand operation logic. To summarize, even if our high-level plan does not include reasoning steps, RAM itself can capture the operational intent and the logic between operations. Therefore, whether integrating reasoning steps into the plan has minimal impact on the results.

### A.6.2 ADDITIONAL DISCUSSION ABOUT RAM

| User Level | Method | CD↓ | | JSD↓ | MMD↓ | COV↑ | F1↑ | | IR↓ |
|---|---|---|---|---|---|---|---|---|---|
| | | Mean | Median | | | | Sketch | Extrusion | |
| Abstract (L0) | w/o RAM | 231.17 | 148.14 | 56.40 | 38.06 | 11.25 | 26.90 | 71.76 | 2.23 |
| | w/ RAM | 208.25 | 127.65 | 44.74 | 22.79 | 33.59 | 35.77 | 87.24 | 2.18 |
| Beginner (L1) | w/o RAM | 254.14 | 188.11 | 44.78 | 32.54 | 13.75 | 39.36 | 83.82 | 3.53 |
| | w/ RAM | 228.62 | 170.15 | 35.98 | 26.99 | 28.12 | 41.77 | 90.81 | 2.75 |
| Intermediate (L2) | w/o RAM | 180.03 | 106.36 | 40.31 | 27.46 | 16.12 | 40.49 | 85.73 | 1.73 |
| | w/ RAM | 158.59 | 77.88 | 16.73 | 16.99 | 31.40 | 53.23 | 92.42 | 1.04 |
| Expert (L3) | w/o RAM | 63.06 | 23.83 | 33.56 | 18.11 | 37.50 | 46.42 | 86.61 | 2.04 |
| | w/ RAM | 34.66 | 0.59 | 4.62 | 5.79 | 66.40 | 57.65 | 92.57 | 2.08 |

Table 9: Additional discussion about RAM.

We also conduct experiments to investigate another question: when the plan includes reasoning steps, is it still necessary to use RAM to extract requirement information from the user instruction? Specifically, using the reasoning plan dataset conducted by OpenAI o3, we perform experiments on two scenarios : (1) training the original PTA with the reasoning plan dataset; (2) training PTA but

Reasoning Plan

1. To fabricate a rectangular prism with a coaxial cylindrical void, we first establish a dedicated coordinate system using the specified Euler angles and translation, ensuring consistent origin and orientation for both features.
2. Next, we sketch a closed rectangle to define the prism's footprint, and add a centrally located circle within the same plane to represent the through-hole.
3. Both profiles are uniformly scaled to achieve the required final size.
4. We then position the scaled sketches in 3D space according to the coordinate system, aligning the sketch plane's normal for proper extrusion. The rectangle is extruded symmetrically to form the solid prism, while the circle is extruded once to create a cylinder matching the prism's thickness.
5. Finally, we subtract the cylinder from the prism, yielding a clean, central cylindrical hole through the rectangular solid.

High-level Plan

1. Create the coordinate system
2. Draw the rectangular for the rectangular prism
3. Draw the circle for the cylindrical hole
4. Scale the 2D sketches
5. Extrude the rectangular and the circle to generate the 3D prism and cylinder
6. subtract the cylinder from the prism

1. To generate a desired tapered cylindrical solid, we first create a new coordinate system with zero Euler rotation and zero translation to serve a common spatial reference for all subsequent operations
2. With that reference frame established, in order to define the cylinder's cross-section, we next open a 2-D sketch on the X-Y plane of this coordinate system.
3. Within that sketch, in order to obtain a closed, watertight profile suitable for solid creation, we draw a single circular face. This circle fully describes the base outline of our cylinder.
4. Having completed the profile, in order to transform it into a three-dimensional body while simultaneously introducing the required slight taper, we extrude the sketch along the positive Z-axis. During the extrusion process, a gentle taper should be created.

1. Create the coordinate system
2. Open a 2D sketch on the X-Y plane
3. Draw the circular face for the cylindrical solid
4. Extrude the sketch to obtain a tapered cylindrical solid

Figure 12: Illustration of the Reasoning plan and the High-level Plan.

discarding the RAM, using only the reasoning plan to guide command sequence generation. The results are provided in Table 9. The results show that the original PTA achieves better performance.

We analyse that user instructions contain comprehensive design concepts and requirement information (e.g., dimensions, geometric relationships). To ensure user satisfaction with the generated CAD models, it is necessary to use the RAM to extract requirements and operational details from the original information provided by the user, even if a plan with a reasoning process is provided.

## A.7 MORE VISUAL RESULTS

Figure 13 presents qualitative comparisons between our PTA and previous state-of-the-art methods. Compared to baseline methods, our PTA accurately perceives the design requirements provided in the user instructions and achieves precise alignment with the user requirements.

## A.8 USAGE OF LLMS

During the preparation of this work, we used OpenAI's ChatGPT (GPT-4 Turbo) in order to translate and improve the author's language. After using this tool, the authors reviewed and edited the content as needed and took full responsibility for the content of the publication.

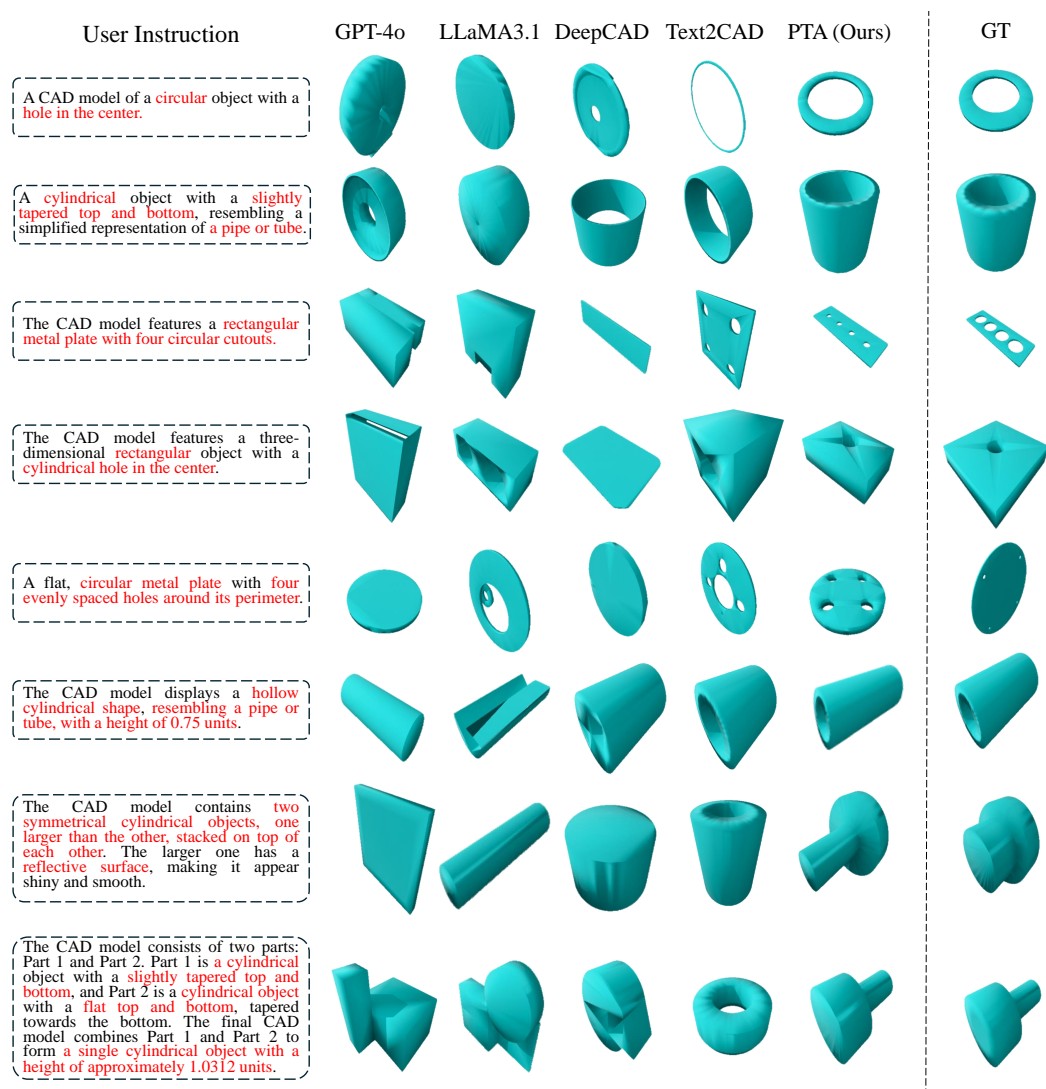

Figure 13: Comparisons with the state-of-the-art methods for CAD generation. Red text indicates information that requires special attention.

