# OpenReview forum: "Plan then Act: Bi-level CAD Command Sequence Generation"
_ICLR.cc/2026/Conference — ICLR 2026 Poster_

### Official Review · Reviewer_VX6V · 2025-10-30

**Soundness:** 2
**Presentation:** 2
**Contribution:** 2
**Rating:** 4
**Confidence:** 4

**Summary:**

This paper proposes a bi-level framework for text-to-CAD command sequence generation, which follows a “Plan then Act” paradigm: an LLM-based Planner first converts user instructions into a structured operation plan, and an Actioner then generates low-level CAD commands guided by extracted design requirements such as dimensions and geometric constraints. Experiments show that PTA achieves superior accuracy and better alignment with user intent compared to existing methods.

**Strengths:**

1. The paper is clear, well-structured, and easy to follow.
2. The proposed method is intuitive and presented concisely.
3. Both the quantitative and qualitative experiments are comprehensive, convincingly demonstrating the effectiveness of the proposed approach.

**Weaknesses:**

1. The method lacks novelty. In fact, the Nli_data dataset already contains more detailed plans, allowing for direct end-to-end generation of Nli_data-format outputs without the need to redesign the planning process.
2. The generation of the high-level plan relies on distilling Qwen2.5-32B-Instruct, whose performance has yet to be thoroughly compared with closed-source models.
3. While Qwen3-8B is a reasoning-focused LLM, the distillation process from Qwen2.5-32B-Instruct does not incorporate intermediate reasoning steps, which may limit its reasoning capabilities.

**Questions:**

Would it be possible to concatenate the user instruction with the high-level plan and use a single text encoder for the subsequent generation?

---

> ### Author Response · Authors · 2025-11-21
>
> Thanks very much for your comments, which are of great significance for us to improve our paper.
>
> > Comment 1: About the novelty of the bi-level generation
>
> **Answer**:
>
> __1). Comparisons between the NLI_data dataset and our High-level Plan.__
>
> When using nli_data with detailed operations as output, the LLM is requested to generate all specific operational steps and precise parameters. Due to LLM hallucinations, this process is highly challenging and often results in incorrect information, ultimately leading to errors in command sequence generation. In contrast, our high-level plan is a chain-like workflow that emphasizes the global operational process without including specific details. The LLM only needs to generate a global plan, which is relatively easier.
>
> To validate the above point, we retrain our method (PTA) using nli_data instead of high-level plan. The experimental results are shown in the table below. It demonstrates that the high-level plan achieves better performance, highlighting its superiority. We have added this analysis to Section 5.3.3 in the revised paper.
> | Plan type        | CD↓ (Mean) | CD↓ (Median) | F1↑ (Sketch) | F1↑ (Extrusion) | IR↓   |
> |------------------|----------|------------|------------|---------------|-------|
> | Nli_data         | 220.61   | 156.00     | 39.88      | 84.85         | 2.46  |
> | **High-level plan**  | **183.14**   | **113.79**     | **50.61**      | **90.33**         | **0.63**  |
>
>
> __2). Regarding the novelty.__
>  In previous end-to-end methods, user instructions are directly mapped to the command sequence without the design of a planning process, resulting in insufficient analysis and utilization of user instruction information. In this paper, we realize the benefits of operation plan and, for the first time, propose a bi-level generation approach:  a high-level plan is generated to serve as operational process guidance, and then user requirements are integrated to produce the command sequence. This new bi-level design enables comprehensive analysis and utilization of the operation and requirement information. The superior performance of our method is demonstrated in Table 1 of the paper.
>
>
> > Comment 2: Comparison between Qwen2.5-32B-Instruct and closed-source LLM.
>
> **Answer**:
> Follow your suggestion, we conduct experiments to compare the performance of Qwen2.5-32B-Instruct and the advanced closed-source LLM OpenAI o3 in generating high-level plans.
> Specifically, we utilize OpenAI o3 to generate high-level plans, resulting in 100,000 triplets of (user instruction, high-level plan, command sequence), and retrain our method(PTA). Similarly, we also retrain PTA using the 100,000 triplets constructed with Qwen2.5-32B-Instruct. The experimental results are provided in the table below. The results show that they achieve comparable performance.  Considering that plan generation involves constructing a complete high-level workflow from detailed operational steps, which is not a difficult task, and Qwen2.5-32B-instruct is capable of accomplishing it.
>
> We have added this analysis to Appendix A.2.2 in the revised paper.
>
> | Source            | CD↓ (Mean)         | CD↓  (Median)       | F1↑ (Sketch)       | F1↑ (Extrusion)       | IR↓   |
> |-------------------|-------------|-------------|------------|------------|-------|
> | Qwen2.5-32B-Instruct | 206.09      | 126.06      | 36.92      | 87.42      | 2.53  |
> | OpenAI o3         | 207.75      | 126.45      | 35.28      | 88.21      | 2.80  |

---

> ### Author Response · Authors · 2025-11-21
>
> > Comment 3: Discussion about incorporating intermediate reasoning steps.
>
> **Answer**:
> This is a highly valuable and thought-provoking point.
> To investigate the impact of intermediate reasoning steps, we conduct experiments to compare the difference between plans with and without the incorporation of reasoning steps. Specifically, we use Qwen2.5-32B-instruct and OpenAI o3 to separately construct plans with reasoning steps (called reasoning plans), each resulting in 100,000 triplets of (user instruction, reasoning plan, command sequence) . These datasets are then used to train PTA, respectively. Similarly, we train PTA using the datasets of 100,000 high-level plans without reasoning steps. The experimental results are provided in the table below. The performance achieved by the reasoning plan is comparable to that of the high-level plan. In other words, integrating reasoning steps into the plan has minimal impact on performance.
>
> | Source          | Plan type | CD↓ (Mean)  | CD↓ (Median)   | F1↑ (Sketch)  | F1↑ (Extrusion) | IR↓ |
> |-----------------|-----------|-------|-------|------|------|-----|
> | Qwen2.5-32B     | Reasoning | 207.01 | 125.03 | 36.33 | 86.35 | 2.61 |
> |     Qwen2.5-32B            | High-level| 206.09 | 126.06 | 36.92 | 87.42 | 2.53 |
> | OpenAI o3       | Reasoning | 208.25 | 127.65 | 35.77 | 87.24 | 2.18 |
> |    OpenAI o3              | High-level| 207.75 | 126.45 | 35.28 | 88.21 | 2.80 |
>
>
> We analyse the interesting results:
>
> The difference between the reasoning plan and the high-level plan is that the reasoning plan includes intermediate reasoning steps, explicitly expressing the operational intent and the logic between operations.
>
> As for our PTA, there is a requirement-aware mechanism (RAM) during the command sequence generation. In the RAM, cross-attention integrates requirement information from the user instruction to obtain operational intent, while self-attention captures contextual relationships to understand operation logic. To summarize, even if our high-level plan does not include reasoning steps, RAM itself can capture the operational intent and the logic between operations. Therefore, whether integrating intermediate reasoning steps into the plan has minimal impact on the results.
>
> We have added this analysis in Appendix A.6.1 of the revised paper.
>
> We also conduct experiments to investigate another question: when a reasoning plan is provided, is it still necessary to introduce RAM? Specifically, we perform experiments on two scenarios : (1) training the original PTA with the reasoning plan dataset conducted by OpenAI o3; (2) training PTA but discarding the RAM, using only the reasoning plan to guide command sequence generation. The experimental results are provided in the table below. The results show that the original PTA achieves better performance.
> | Method | CD↓ (Mean)  | CD↓ (Median)   | F1↑ (Sketch) | F1↑ (Extrusion) | IR↓ |
> |--------|-------|-------|------|------|-----|
> | w/o RAM | 231.17 | 148.14 | 26.90 | 71.76 | 2.23 |
> | **w/ RAM**  | **208.25** | **127.65** | **35.77** | **87.24** | **2.18** |
>
> We analyse that user instructions contain the most comprehensive design concepts and requirement information (e.g., dimensions, geometric relationships). To ensure user satisfaction with the generated CAD models, it is necessary to use the RAM to extract requirements and operational details from the original information provided by the user, even if a plan with a complete reasoning process is provided.
>
> We have added this analysis to Appendix A.6.2 in the revised paper.

---

> ### Author Response · Authors · 2025-11-23
>
> > Question: Concatenation of the user instruction with the high-level plan.
>
> **Answer**:
> Yes, concatenating the user instruction with the high-level plan and feeding them into a single encoder is a straightforward way of fusing information.
>
> As for our PTA, we introduce a requirement-aware mechanism(RAM). The RAM  automatically extracts the relevant design requirements from the user instruction for each operation in the plan and captures the logical relationships between the operations. In our ablation studies (cf. Table 3 of the main paper), we compare RAM with feature-level concatenation and addition, and the results validate the effectiveness of our RAM.
>
> Follow your question, we conduct experiments where the user instruction and high-level plan are concatenated as input to a single text encoder. The results are provided in the table below. Concatenation is inferior to our designed requirement-aware mechanism(RAM). This demonstrates that our RAM more effectively achieves information fusion between the high-level plan and the user instruction.
>
> | Guidance in Actioner | CD↓ (Mean)  | CD↓ (Median)  | F1↑ (Sketch) | F1↑ (Extrusion) | IR↓ |
> |---------------------|-------|-------|------|------|-----|
> | Concatenation       | 198.09 | 160.71 | 45.99 | 80.65 | 2.77 |
> | **RAM (Ours)**          | **183.14** | **113.79** | **50.61** | **90.33** | **0.63** |
>
> ***
>
> We sincerely appreciate your valuable and thoughtful feedback on our manuscript. We have carefully addressed each of your comments and incorporated the necessary revisions into the official manuscript version. We hope that our revisions meet your expectations and would be truly grateful if you could consider revising the rating based on the improvements made. We welcome any further questions or discussions you may have, and we will be happy to provide more elaborate responses as needed！

---

### Official Review · Reviewer_zpuM · 2025-10-30

**Soundness:** 3
**Presentation:** 3
**Contribution:** 3
**Rating:** 6
**Confidence:** 3

**Summary:**

This work provides a novel framework, Plan then Act (PTA), for text-to-CAD synthesis, where given a user instruction on the 3D CAD model to construct, the pipeline consists of a “planner” that first converts the instructions to a detailed step-by-step guide and an “actioner” which takes in both the original instruction and the plan to output the low-level CAD construction commands. The planner is implemented as a LLM that  is finetuned on high-level plans extracted from Text2CAD [1].  The actioner consists of BERT text encoders followed by a cross-attention based mechanism (named “Requirement-Aware Mechanism”) to fuse the representations of the user instruction and the high-level plans, from which the construction commands are decoded. Experiments demonstrate that on the Text2CAD test set, the proposed method achieves enhanced performance compared to LLMs that directly output CAD commands across 7 metrics.

**Strengths:**

1. The approach is novel and the motivation of a two-stage approach rather than directly generating the low-level construction commands is clear.
2. Experiment results show a significant improvement across different levels of detail in user prompts compared to open-source LLMs trained on the same dataset.
3. Ablation experiments are thorough in showing the contributions of each stage in the pipeline.

**Weaknesses:**

1. While the work compares to multiple open-source LLM baselines, there is only one closed-source model evaluated. It would be good to include reasoning models (e.g. o3 or o4), and stronger closed-source LLMs like Gemini-2.5-Pro and GPT-5.
2. It is somewhat unclear why the construction commands need to be decoded from a fused representation of the plans and the user instruction instead of just the high-level plans. This makes me wonder is there some information loss when generating the plans?

**Questions:**

1. What is the performance of more recent closed source LLMs on Text2CAD?
2. I am unsure about the necessity of the RAM compared to decoding from the high-level plans. Additional explanation and examples in the text could be beneficial.
3. What is the performance of training PTA end-to-end, without directly supervising the high-level plans?

---

> ### Author Response · Authors · 2025-11-24
>
> Thanks for your positive feedback and valuable questions. We greatly appreciate your recognition of the novelty and contributions of our method.  In the following, we will respond to your questions one by one.
>
> > Comment 1 & Question 1:  Comparisons with more closed-source LLMs.
>
> **Answer**:  Thanks for your question. We compared our PTA with GPT-4o, a representative closed-source LLM. Since GPT-4o exhibits a high invalid rate in generating CAD command sequences, we consider that it represents the performance of closed-source LLMs on this task.
>
> Following your suggestion, we compare our  PTA with OpenAI o3, OpenAI o4-mini, Gemini-2.5-Pro, and GPT-5. OpenAI o3 and o4-mini are reasoning LLMs, while Gemini-2.5-Pro and GPT-5 are more powerful closed-source LLMs. We regret that OpenAI has not made the o4 API publicly available,
> so we perform experiments on OpenAI o4-mini instead.
> The experimental results are shown in the table below. Similar to GPT-4o, these LLMs also yield a high invalid rate (IR), with most generated command sequences failing to be correctly executed and converted into CAD models. These results demonstrate that, compared with closed-source LLMs, our PTA achieves superior performance in CAD command sequence generation.
> We have added these results to Appendix A.4 in the revised paper.
>
> | Method            | CD↓ (Mean)         | CD↓  (Median)       | F1↑ (Sketch)       | F1↑ (Extrusion)       | IR↓   |
> |-------------------|-------------|-------------|------------|------------|-------|
> | GPT-4o          | 338.52  | 247.11    | 16.12  | 63.51        | 70.35|
> | OpenAI o3      | 312.99  | 163.99    | 26.17  | 61.65        | 68.09|
> | OpenAI o4-mini | 284.12  | 218.36    | 25.67  | 55.78        | 77.25|
> | GPT-5          | 269.73  | 197.63    | 37.06  | 64.84        | 67.99|
> | Gemini-2.5-Pro | 237.78  | 186.79    | 35.75  | 66.31        | 85.77|
> | **PTA (Ours)**     | **183.14**  | **113.79**    | **50.61**  | **90.33**        | **0.63** |
>
>
> > Comment 2 & Question 2: Why use RAM to fuse plans and the user instructions?
>
> **Answer**:
> Previous end-to-end methods struggle to directly map user instructions to complex low-level command sequences due to limited semantic parsing or command sequence modeling capabilities (cf. Lines 39-47).
>
> In contrast, our PTA first constructs a high-level, chain-like operations plan that provides global operation guidance for the command sequence generation. While this high-level plan is derived from the user instruction, to generate an accurate and precise command sequence, it is necessary to integrate the high-level plan with the specific operational details (cf. Table 3). To achieve this, we introduce the requirement-aware mechanism (RAM). Specifically, for each operation step in the high-level plan, RAM identifies and fuses the relevant design requirements from the user instruction. The fused representation of operational steps and design requirements effectively supplements the operational details and jointly guides the command sequence generation.
>
> Through our bi-level method: first constructing the high-level plan and then capturing operational details, our PTA enables more comprehensive analysis and utilization of both operational and requirement information, ultimately leading to more accurate command-sequence generation (cf. Table 1).
>
> > Question 3: The performance of the end-to-end trained PTA.
>
> **Answer**:  Thanks for your question. Our current PTA can not support end-to-end training. In PTA, we first use an LLM-based Planner to parse the user instruction into a high-level plan in text format, and then employ a designed Actioner to generate the command sequence. The loss from the output command sequence cannot be backpropagated through the discrete text-format plan to the Planner.
> Following your valuable suggestion, we will actively explore end-to-end training implementation in our future work.
>
> ***
> Thank you again for your insightful suggestions to improve our paper!  We hope that our responses effectively answer your questions.

---

### Official Review · Reviewer_oPg7 · 2025-11-01

**Soundness:** 3
**Presentation:** 3
**Contribution:** 3
**Rating:** 6
**Confidence:** 3

**Summary:**

This paper presents a two-tier approach for natural-language based CAD modeling: an LLM-based Planner is first asked to create a high-level plan from user instructions, then a separate Actioner formulates a low-level command sequence executing the information given by the combined and strategically attenuated Plan + original user instruction. The combined features are driven by the Requirement-Aware mechanism introduced in this paper, which (1) identifies contextual relationships among the steps of the Plan, and (2) identifies complementary information between the high-level Plan and the original user-specified requirements. The authors provide extensive baseline comparisons and ablations, showing excellent relative performance of the proposed PTA approach.

**Strengths:**

The effective application of this approach to CAD is exciting, particularly with the commanding advantage that this method displays over other state of the art solutions, with respect to both quantitative and qualitative metrics. The overarching principles (multi-step reasoning/inference/generation, multi-technology approaches, reduction of user burden) are also valuable independent of the domain, and they are likely to generalize well beyond CAD. The paper and the method within are well motivated, and presented clearly. The ablation studies effectively demonstrate the contribution of each piece of the approach.

**Weaknesses:**

1. The main paper is currently lacking a suitable description for the dataset used to finetune the Planner. For example, how different are the nli_data files and the target high level plan that you seek from Qwen? How do you verify the quality/correctness of the Qwen output before using that as the basis for your Planner training? I realize some of this is already in appendix A2, but since it is an important piece of your approach, an additional few sentences in the main text would be very helpful.

2. I am somewhat unsatisfied/unconvinced by the need for the convoluted RAM and joint F_plan + F_inst representation. To be clear, the ablation studies and empirical evidence in the left side of Figure 2 have convinced me that the RAM and joint representation are necessary _given the current structure of the plan_. However, I'm curious what precludes the generation of a high-level plan that is complete and self-contained, such that the instructions are redundant? This is also tied to my previous question about the dataset for Planner training, as an improved Plan quality could dramatically simplify the process without sacrificing performance. Can the authors comment on whether a more complete Plan might be feasible, what the challenges are, and/or why the current approach is preferable/necessary?

3. There is minimal discussion about the limitations and assumptions of this system; the paper would benefit from such a discussion.

4. The current structure of Section 4 is somewhat disjointed and hard to read, thanks to the multiple levels of overviews.  For example, at l.191, the description is sufficiently low level that I found myself wondering about/looking for the details of L_plan, only to find that it would be deferred to l. 232. Similarly, at l.227, I wondered about the specific dataset/training procedure used to train the Qwen model (which are only detailed through Section 5.1, and Appendix). It may be worth another writing pass to smooth out this experience and connect the ideas more directly. Specifically, I would make the overviews slightly higher level, and add sufficient detail to the actual explanation as necessary.

**Questions:**

1. In lines 153-157, why are the Invalid Rates suddenly so much better? I'm particularly curious for Case 1, which seems identical to the experiment in the previous paragraph (where much higher invalid rates were reported)? Could you clarify what is different?


## Minor points
The paper contains many typos and grammatical errors (some listed below), which are occasionally severe enough to obscure the meaning. Please take another careful proofread.

- l.49-51 -- should that quote have a citation, or are the quotes just to indicate your assumption (in which case they could be removed)?
- l. 79, 81, 239, throughout -- missing articles (<the> operation steps in <the> high level plan... guides <a> Transformer decoder) or verbs (our goal <is> to efficiently utilize)
- l. 156-157 -- metrics like CD and F1 should be spelled out, given some additional context, or deferred to later when they're properly introduced.
- l. 148 -- how were LLaMA and Qwen finetuned for this experiment? Could you include details about the process/prompts/data/parameters to contextualize the experiment, either in the main text or in the appendix (with a forward reference)?
- l. 244-245 -- clarify the dimensions, e.g. N_t, N_p, d_p. They don't appear in Figure 2 or the text, as far as I saw.
- l.311 - Is there a reason that CAD-GPT would be a uniquely suitable dataset in your case? If not, it seems odd to mention it purely to say that it's unavailable; I would consider justifying or removing that line.
- l. 366, 367 -- the description of JSD is unclear to me. The description of COV has grammatical issues that preclude understanding.
- l. 466 -- efficiency --> efficacy
- l. 473 -- absence of <required> information
- l. 474 -- specially --> (specifically? especially?); onformation --> information

---

> ### Author Response · Authors · 2025-11-23
>
> Thanks for your careful and positive feedback. We greatly appreciate your recognition of our motivation and contributions. In the following, we will respond to your questions one by one.
>
> > Comment 1: Lack of a suitable description for the high-level plan dataset
>
> **Answer**:
>
> __1).Difference between nli_data and high-level plan.__
>
> The nli_data is detailed low-level operational action sequences with precise parameters in natural-language form. In contrast, the high-level plan emphasizes the overall operational process without including specific details or parameters.
> We have clarified the differences between nli_data and high-level plan in the revised paper (Sec. 5.3.3).
>
>
> __2). Evaluation of High-level Plan Quality.__
>
> In fact, during data construction, we manually sample and check the high-level plans generated by Qwen2.5-32B-Instruct and find that they are of high quality to meet our experimental requirements. To further support this assessment, we conduct a user study and quantitative experiments to evaluate the quality of the high-level plan.
>
> 1. **User study**: We randomly select 1,000 high-level plan samples, which are evenly divided into five groups. We then invite 30 volunteers with at least a bachelor’s degree, and every volunteer is assigned to one data group. For each sample, participants are presented with the user instruction and  nli_data, and then asked to judge whether the corresponding high-level plan is satisfactory (binary classification: any unreasonable aspect will be counted as unsatisfactory). In total, we collected 6,000 responses, with a satisfaction rate of 98.75%.
>
> 2. **Quantitative experiments**: we compare Qwen2.5-32B-Instruct with the advanced reasoning LLM OpenAI o3, demonstrating Qwen2.5-32B-Instruct’s ability to generate satisfied plans. Specifically, we employ OpenAI o3 to generate high-level plans and resulting 100,000 triplets of (user instruction, high-level plan, command sequence) . Using this dataset, we retrain our PTA. Similarly, we also train PTA using the 100,000 triplets constructed with Qwen2.5-32B-Instruct.  The experimental results are provided in the table below. The results show that they achieve comparable performance. Considering that plan generation involves constructing a high-level workflow from detailed operational steps, which is not a difficult task, and Qwen2.5-32B-instruct is capable of accomplishing it.
>
> | Source            | CD↓ (Mean)         | CD↓  (Median)       | F1↑ (Sketch)       | F1↑ (Extrusion)       | IR↓   |
> |-------------------|-------------|-------------|------------|------------|-------|
> | Qwen2.5-32B-Instruct | 206.09      | 126.06      | 36.92      | 87.42      | 2.53  |
> | OpenAI o3         | 207.75      | 126.45      | 35.28      | 88.21      | 2.80  |
>
> In summary, we consider that the high-level plans generated by Qwen2.5-32B-Instruct are of high quality and can be used to train our method.
> We have added these discussions to the revised paper (Supp. A.2.2).
>
> > Comment 2: More complete plan
>
> **Answer**:
> Thanks for your question. The user instructions contain both the overall design intent and comprehensive design requirements (e.g., dimensions, geometric relationships). To ensure user satisfaction with the generated CAD models, it remains necessary to supplement the plan with information from the user instructions during command sequence generation, even with a complete plan.
>
> To validate the above point, we also design a more complete plan with reasoning steps (called reasoning plans), which include explicit operational intents and logical relationships between steps. We investigate whether, given such a reasoning plan, it is still necessary to extract information from the user instruction again during command-sequence generation.
>
> Specifically, using OpenAI o3, we conduct 100,000 triplets of (user instruction, reasoning plan, command sequence). Based on this dataset, we perform experiments under two scenarios: (1) training the original PTA; (2) retraining PTA while discarding user instructions during command sequence generation, using only the reasoning plan as guidance. The experimental results are provided in the table below. When using only the plan as guiding information (w/o user instruction), there is a performance significant deterioration, indicating that it is necessary to fuse design requirements from the user instructions and reinforce the design intent during command-sequence generation. We consider that our RAM enables a more comprehensive analysis and utilization of both operational and requirement information, resulting in accurate command sequence generation.
> We have added these discussions to Supp. A.6.2 in the revised paper.
>
> | Method | CD↓ (Mean)  | CD↓ (Median)   | F1↑ (Sketch) | F1↑ (Extrusion) | IR↓ |
> |--------|-------|-------|------|------|-----|
> | w/o user instruction | 231.17 | 148.14 | 26.90 | 71.76 | 2.23 |
> | **w/ user instruction**  | **208.25** | **127.65** | **35.77** | **87.24** | **2.18** |

---

> ### Author Response · Authors · 2025-11-23
>
> > Comment 3: Discussion about limitation.
>
> **Answer**:
> Thanks for your suggestion.
> We provide a failure case analysis in Appendix A.5; to make it more directly visible to readers, this has been moved to the main paper in the revised version (cf. Section 5.3.4).
> Specifically, when user instructions contain descriptions that are uncommon in the training set, our PTA struggles to generate CAD models that fully meet user requirements. A simple solution to address this issue is to perform data augmentation for rare descriptions during training.
>
> > Comment 4: Disjointed structure of Section 4
>
> **Answer**:
> Thank you for pointing out the writing issues.
> Following your suggestion, in the revised paper, we have rewritten Section 4: the overview is now more high-level, and the explanation part includes more details. Specifically, we no longer mention the details such as L_plan in the overview. We have also added details of the dataset and training process in Section 4.2.2.
>
> > Question: Different invalid rates.
>
> **Answer**:
> Thanks for your question. We apologize for the unclear description. The mentioned experiments are conducted on different model architectures, thereby achieving different invalid rates.
>
> More specifically, the experiment in lines 153–157 is designed to evaluate the effectiveness of the operation plan and is conducted on a CAD generation model trained from scratch. Differently, the experiment in the previous paragraph examines the ability of LLMs to directly generate command sequences, using LLMs (GPT-4o, LLaMA3.1, Qwen3). We have clarified this distinction in Section 3 of the revised paper.
>
> >  Minor points
>
> **Answer**:
> Thanks for your correction; we apologize for the minor issues.
> - The quotation marks have been removed (Lines 49-51 in the revised paper).
> - We corrected the grammatical errors (Lines 79-81 in the revised paper).
> - We added explanations of the metrics in Section 3.
> - We have included details of fine-tuning LLaMA3.1 and Qwen3 in Appendix A.4 of the revised paper.
> - We clarified the meanings of the symbols (Lines 248-249 in the revised paper).
> - The CAD-GPT dataset is not open-source, so we could not use it; we have clarified this in Section 5.1.1 of the revised paper.
> - We corrected the descriptions of the metrics JSD and COV in Section 5.1.2.
> - We fixed word errors (line 474 in the revised paper).
> - We corrected grammatical mistakes (Line 481 in the revised paper).
> - We corrected spelling errors (Lines 482-483 in the revised paper).
>
> ***
>
> Hope that our responses address your questions. If you have any remaining inquiries, please let us know!

---

### Official Review · Reviewer_3FuS · 2025-11-02

**Soundness:** 3
**Presentation:** 2
**Contribution:** 2
**Rating:** 6
**Confidence:** 3

**Summary:**

This paper proposes a method for generating CAD command sequences from natural language text. The authors first discuss how pre-trained large language models (LLMs) on general large-scale data struggle to directly generate task-specific, low-level CAD control commands. They then note that operation plans can be helpful in improving the accuracy of generating CAD command sequences. Based on these observations, they design a two-stage approach: first, using an LLM to generate a plan from the user text, and then employing a cross-attention network to combine these two text encodings to output the CAD command sequence.

**Strengths:**

1. The motivation for a two-stage approach—first generating a plan and then generating the command sequences—is interesting.
1. The discussion of directly generating command sequences and the high failure rates associated with this approach provides reasonable grounds for the further use of a two-stage process.
1. The experimental results appear convincing, showing that the method is useful.

**Weaknesses:**

1. Direct generation of command sequences with LLMs had higher failure rates, both with prompting and finetuning, although finetuning helped, as mentioned. Did the authors further try other post-training approaches, such as reinforcement learning?
1. It is not clear why cross-attention between the text and plan is useful. The plan should already be sufficient to generate reasonable command sequences, since it is derived from the user text and is more structured.

**Questions:**

Please see weaknesses.

---

> ### Author Response · Authors · 2025-11-24
>
> Thank you for your positive feedback and valuable suggestions. We greatly appreciate your recognition of the motivation and value of our method. In the following, we will respond to your questions one by one.
>
> > Comment 1: Other post-training approaches, like reinforcement learning.
>
> **Answer**:
> When adapting general LLMs to a specialized domain, it is common to use supervised finetuning (SFT) [a][b] or SFT followed by reinforcement learning (RL) [c]. Considering the significant differences between CAD command sequences and general-purpose data, SFT is necessary for the general LLMs. In our studies, supervised finetuning of both LLaMA3.1-8B and Qwen3-8B led to a high invalid rate. Based on these results, we consider that applying RL on such poorly finetuned LLMs is unlikely to yield substantial improvements.
>
> To validate the above point, we apply direct preference optimization (DPO), a reinforcement learning technique, to the supervised-finetuned Qwen3-8B. As shown in the table below, the model still produces a relatively high invalid rate, which demonstrates the difficulty for LLMs to directly generate command sequences.
> We have added these discussions to Appendix A.4 in the revised paper.
>
> | Method            | CD↓ (Mean)         | CD↓  (Median)       | F1↑ (Sketch)       | F1↑ (Extrusion)       | IR↓   |
> |-------------------|-------------|-------------|------------|------------|-------|
> | Qwen3-8B (Only supervised finetuning)| 248.93      | 218.97      | 31.92      | 71.73      | 20.06  |
> | Qwen3-8B (Supervised finetuning + DPO)| 226.94      | 193.07      | 33.86      | 76.20      | 18.29  |
> | **PTA (Ours)**         | **183.14**      | **113.79**      | **50.61**      | **90.33**      | **0.63**  |
>
>
> [a] Interpretable long-form legal question answering with retrieval-augmented large language models. AAAI 2024
>
> [b] Enabling Few-Shot Alzheimer's Disease Diagnosis on Tabular Biomarker Data with LLMs. Arxiv 2025
>
> [c]  CareBot: A Pioneering Full-Process Open-Source Medical Language Model. AAAI 2025
>
>
>
> > Comment 2: Why fuse text and plan?
>
> **Answer**:
> Previous end-to-end methods struggle to directly map user instructions to complex low-level command sequences due to limited semantic parsing or command sequence modeling capabilities ( cf. Lines 39-47).
>
> In contrast, our PTA first constructs a high-level, chain-like operations plan that provides global operation guidance for the command sequence generation. While this high-level plan is derived from the user instruction and is well-structured, to generate an accurate and precise command sequence, it is necessary to integrate the high-level plan with the specific operational details ( cf.  Table 3).
>
> To achieve this, we introduce the requirement-aware mechanism (RAM).  Specifically, for each operation step in the high-level plan, RAM automatically extracts the most relevant design requirements (e.g., dimensions, geometric relationships) from the user instruction, thereby effectively supplementing the operational details and producing an accurate command sequence.
> Through our bi-level method: first constructing the high-level plan and then capturing operational details, our PTA enables more comprehensive analysis and utilization of both operational and requirement information, ultimately leading to more accurate command-sequence generation ( cf. Table 1).
>
> ***
> Thanks again for your valuable suggestions! I hope our responses effectively answer your questions.

---

### Public Comment · ~Pacheco1 · 2025-11-28
**Some suggestions for authors and reviewers**

Dear Authors and Reviewers,

As an expert in CAD with prior publications at CVPR, ICCV, and related venues, I would like to provide several suggestions on this work.

> Methodology:

- **Lack of architectural innovation**: The model architecture closely follows Text2CAD, with no significant architectural innovation apparent. In my view, this work appears to be an extension of Text2CAD.

- The main contribution lies in the Planner, which converts user inputs into a plan-like format. Is this transformation truly beneficial, even if numerical results improve slightly? Moreover, as shown in Appendix Figure 6, your method appears to be a highly condensed version of the Text2CAD data. Can you guarantee the accuracy of the data description?

> Data:

**Outdated data representation**:
- The authors adopt the same representation used in Text2CAD, which is a serialized format. This form lacks precise controllability and editability, and has already been abandoned in the CAD community. Recent state-of-the-art efforts focus on CAD code or LLM-based representations, which enable industrial-level control and are thus more suitable for modern CAD applications.
- The text dataset is also built upon Text2CAD, with no contribution to new data.

> Results:
- **Unfair experimental comparison**: Are the performance improvements in Table 1 due to the Planner’s preprocessing of the input text? Would it be fairer to run all methods using the Planner’s output as input?

- **Insufficient comparison**: As shown in Table 1, the most recent baseline included is only Text2CAD (NeurIPS’24). To the best of my knowledge, several recent CAD-related works have already been released, such as Drawing2CAD (MM’25), CAD-Editor (ICML’25), and GenCAD (TMLR’25). Although their data representations may differ, fair quantitative comparisons are still possible since they are all built on the DeepCAD dataset. The proposed method does not demonstrate sufficient capability to achieve state-of-the-art performance.
- **Overly simple generated results**: Current CAD generation methods can produce fine-grained and complex structures, while the results in this paper are mostly limited to simple shapes such as cylinders and rectangles. The outputs appear too simplistic and lack controllability.
- **Effectiveness of RAM?** Is a simple attention mechanism truly effective? Even though you compare Add and Concat in ablation studies, I remain skeptical about the dataset's reliability.
- **Lack of cross-domain experiments**: There is no evidence to show whether the model generalizes beyond the current data distribution or merely overfits it.

In conclusion, although the authors report decent scores, there remains substantial room for improvement both in terms of technical innovation and presentation.

---

> ### Author Response · Authors · 2025-12-03
>
> Thanks for your attention to our paper. We will respond to these questions one by one.
>
> > Methodology
>
> **Answer**:
>
> 1). Regarding architectural innovation.
>
> Our bi-level generation method is fundamentally different from Text2CAD. Text2CAD is an end-to-end generation method that encodes text instructions with an encoder and feeds the text features into a Transformer decoder to produce the CAD command sequence.  It struggles to directly map user instructions to complex low-level command sequences due to its limited semantic parsing capabilities (cf. lines 41-43).
>
> Unlike Text2CAD, we propose a new bi-level generation method for the first time:  it produces a complete high-level operation plan and then integrates design requirements to supplement the operational details, ultimately producing an accurate low-level executable CAD command sequence.  This bi-level design enables a more comprehensive analysis and utilization of both operational and requirement information. The experimental results (cf. Table 1) demonstrate that our PTA achieves significant performance improvements over Text2CAD.
>
> In our method, we use the same text encoder and Transformer decoder as Text2CAD, but these are merely widely-used implementation components rather than contributions of our work.  The core contribution of our paper lies in the proposed bi-level generation, and it has been unanimously recognized by the official expert reviewers (3FuS, oPg7, zpuM).
>
> 2). In our bi-level generation (Plan then Act), the Planner is responsible for generating a high-level operation plan based on the user instruction.  This high-level plan provides complete operation process guidance for the generation of the CAD command sequence.
> We provide extensive experimental results (cf. Table1-2, Figure 4, and Figure 13) to demonstrate the benefit of bi-level generation.  The effectiveness of this bi-level generation has been recognized by the official expert reviewers (3FuS, oPg7, zpuM, VX6V).
>
> Appendix Figure 6 (Appendix Figure 7 in the revised paper) illustrates the process of extracting high-level plans from the nli_data of the Text2CAD dataset.  These high-level plans are used as training data for our bi-level generation method.   To evaluate the quality of the constructed plans, we perform a user study and a quantitative comparison experiment, and the results(cf. Appendix A.2.2 in the revised paper) confirm the plan's accuracy.
>
> > Concerns about data representation.
>
> **Answer**:
> - We use the widely used CAD command sequence representation[A-F], which is also employed by Text2CAD. The CAD command sequence allows designers to control the design history and iteratively refine CAD models, retaining full controllability and editability [A-C]. This representation remains valuable for research and is still widely used, as evidenced by recent studies such as Drawing2CAD (MM’25) [D], CAD-Editor (ICML’25) [E], and GenCAD (TMLR’25) [F], all of which also adopt CAD command sequences as their data representation.
> - To train our bi-level generation method, we constructed high-level plan data based on the Text2CAD dataset. This plan data serves only as the data foundation for our method, and we never claimed it serves as a contribution (cf. Lines 85-94).

---

> ### Author Response · Authors · 2025-12-03
>
> > Results.
>
> **Answer**:
>
> **Clarification of experimental comparison in  Table 1.**
>
> 1). The experimental comparisons in Table 1 are entirely fair. All methods except the closed-source LLM GPT-4o are trained and evaluated on the Text2CAD dataset. We also explicitly emphasized the fairness of these comparisons in Section 5.1.4.
>
> It is important to emphasize that the Planner is not a preprocessing module; rather, it is an essential component of our bi-level generation method. The Planner is responsible for generating a complete high-level operation plan based on the user instruction. This high-level plan provides complete operation process guidance for generating the low-level CAD command sequence. The performance improvements it brings are precisely part of the advantages of our method (cf. Table 2 ).  Even if using the planner’s outputs may improve the performance of other methods, this would further demonstrate the effectiveness of our bi-level generation. Inspired by your suggestion, we will consider developing the planner as a plug-and-play module in future work.
>
> 2). We sufficiently compare our method (PTA) with advanced closed-source LLM(GPT-4o), open-source LLM (LLaMA3.1), and the available state-of-the-art Text-to-CAD methods (Text2CAD) in Table 1. The experimental results demonstrate the superior performance of our method. The mentioned methods, such as Drawing2CAD (MM’25), which generates CAD models from 2D engineering drawings; CAD-Editor (ICML’25), which focuses on CAD editing; and GenCAD (TMLR’25), which generates CAD from images. They address fundamentally different tasks from ours, which focus on generating CAD from text instructions. Therefore, even though they are recent CAD-related works, it is not reasonable to compare methods designed for different tasks, and we do not include them as baselines.
>
> **Clarification about generating complex CAD.**
>
> We present complex structures such as screw caps, desks, and multi-object assemblies in Figure 4 and Figure 13, rather than only simple shapes such as cylinders and rectangles. Moreover, official reviewer 3FuS considered our experimental results convincing, and reviewers oPg7 and VX6V also expressed their recognition of our qualitative results.
>
>
> **Motivation and effectiveness of RAM.**
>
> Our bi-level generation method(PTA) first constructs a high-level, chain-like operations plan that provides global operation guidance for the command sequence generation. While this high-level plan is derived from the user instruction, to generate an accurate and precise command sequence, it is necessary to integrate the high-level plan with the specific operational details (cf. Table 3).
>
> To achieve this, we introduce the requirement-aware mechanism (RAM). Specifically, for each operation step in the high-level plan, RAM identifies and fuses the relevant design requirements from the user instruction. The fused representation of operational steps and design requirements effectively supplements the operational details and jointly guides the generation of the command sequence. The experiments (cf. Table 3 and Table 9 in the revised paper) validate the effectiveness of RAM.
>
>
> **Concerns about overfitting.**
>
> Following the prior SOTA method[A], we use the large-scale Text2CAD dataset, which contains 660k samples, and we adhere to a strict data split of 600k for training, 32k for validation, and 32k for testing. This large-scale dataset and non-overlapping split enable selecting the model with superior generalization performance. Moreover, the effectiveness of our method has already been acknowledged by the official reviewers (3FuS, oPg7, zpuM, VX6V).
>
> [A] Text2CAD: Generating Sequential CAD Models from Beginner-to-Expert Level Text Prompts. NIPS 2024
>
> [B] DeepCAD: A Deep Generative Network for Computer-Aided Design Models. ICCV 2021
>
> [C] CAD-GPT: Synthesising CAD Construction Sequence with Spatial Reasoning-Enhanced Multimodal LLMs. AAAI 2025
>
> [D] Drawing2CAD: Sequence-to-Sequence Learning for CAD Generation from Vector Drawings. ACM MM 2025
>
> [E] Cad-editor: A locate-then-infill framework with automated training data synthesis for text-based cad editing. ICML 2025
>
> [F] Gencad: Image-conditioned computer-aided design generation with transformer-based contrastive representation and diffusion priors. TMLR 2025

---

### Author Response · Authors · 2025-12-03
**The summary of rebuttal**

Dear AC and reviewers,

In this section, we aim to highlight the contributions of our work and summarize the key points from our rebuttal. Our study focuses on the task of CAD generation conditioned by text instructions.To achieve accurate CAD modeling, we propose a new bi-level generation method (PTA): a complete high-level operation plan is first produced based on user instructions, and user requirements are then incorporated to fill in the operational details, ultimately generating an accurate low-level executable CAD command sequence.  Our bi-level generation method enables more comprehensive analysis and utilization of semantic information (cf. Table 2), resulting in more accurate command sequence generation (cf. Table 1 and Figure 4).

We are pleased that the reviewers (3FuS, oPg7, zpuM) recognized the motivation and contributions of our bi-level generation method, and that reviewers (3FuS, oPg7, zpuM, VX6V) acknowledged the comprehensiveness of our quantitative and qualitative experiments.   During rebuttal, we have provided detailed responses to the reviewers’ questions, mainly covering:

- Comparisons with more closed-sourced LLMs;
- The necessity of introducing RAM to fuse the plan and instruction;
- Evaluation of the conducted high-level plan quality;
- The impact of introducing reasoning steps into the high-level plan on performance.

We hope that these revisions adequately address all the reviewers' concerns and further strengthen the contributions of our study.  We sincerely appreciate the time and effort that the reviewers have dedicated to evaluating our paper.    Their insightful comments and constructive feedback have significantly improved the quality of our work.    We are also grateful for the AC's efforts throughout the rebuttal process.

Warm regards,

The Authors.

---

### Meta-Review · Area_Chair_hqSG · 2026-01-07

**Summary:**

This paper proposes PTA, a novel bi-level method for CAD command sequence generation. PTA follows a “Plan then Act” paradigm: an LLM-based Planner first converts user instructions into a structured operation plan, and an Actioner then generates low-level CAD commands guided by extracted design requirements, such as dimensions and geometric constraints. Experimental results demonstrate that PTA outperforms existing methods in both quantitative metrics and qualitative evaluations

**Reviewer Concerns:**

During the rebuttal, the authors added more experiments on both open- and closed-source models, as well as experiments analyzing reasoning steps. The main concerns remain the novelty of the architecture and the limited baseline comparisons. The work could be strengthened further by providing additional experiments that isolate the contribution of each architectural design and demonstrate its effectiveness across a wider range of models.

**Reviewer Scores:**

The rebuttal addresses some concerns and may warrant a slight increase in confidence. Overall, given the remaining issues, I consider this paper to be borderline.

---

### Decision · Program_Chairs · 2026-01-26

Accept (Poster)